# Tempo and mode of gene expression evolution in the brain across primates

Katherine Rickelton[1,2]*, Trisha M Zintel[1,2], Jason Pizzollo[1,2], Emily Miller[1], John J Ely[3,4], Mary Ann Raghanti[5], William D Hopkins[6], Patrick R Hof[7,8], Chet C Sherwood[3], Amy L Bauernfeind[9,10], Courtney C Babbitt[1]*

[1]Department of Biology, University of Massachusetts Amherst, Amherst, United States; [2]Molecular and Cellular Biology Graduate Program, University of Massachusetts Amherst, Amherst, United States; [3]Department of Anthropology and Center for the Advanced Study of Human Paleobiology, The George Washington University, Washington, United States; [4]MAEBIOS Epidemiology Unit, Alamogordo, United States; [5]Department of Anthropology, School of Biomedical Sciences, and Brain Health Research Institute, Kent State University, Kent, United States; [6]Department of Comparative Medicine, Michale E. Keeling Center for Comparative Medicine,The University of Texas M D Anderson Cancer Centre, Bastrop, United States; [7]New York Consortium in Evolutionary Primatology, New York, United States; [8]Nash Family Department of Neuroscience and Friedman Brain Institute, Icahn School of Medicine at Mount Sinai, New York, United States; [9]Department of Neuroscience, Washington University School of Medicine, St. Louis, United States; [10]Department of Anthropology, Washington University in St. Louis, St. Louis, United States

*For correspondence:
krickelton@umass.edu (KR);
cbabbitt@bio.umass.edu (CCB)

**Competing interest:** The authors declare that no competing interests exist.

**Abstract** Primate evolution has led to a remarkable diversity of behavioral specializations and pronounced brain size variation among species (Barton, 2012; DeCasien and Higham, 2019; Powell et al., 2017). Gene expression provides a promising opportunity for studying the molecular basis of brain evolution, but it has been explored in very few primate species to date (e.g. Khaitovich et al., 2005; Khrameeva et al., 2020; Ma et al., 2022; Somel et al., 2009). To understand the landscape of gene expression evolution across the primate lineage, we generated and analyzed RNA-seq data from four brain regions in an unprecedented eighteen species. Here, we show a remarkable level of variation in gene expression among hominid species, including humans and chimpanzees, despite their relatively recent divergence time from other primates. We found that individual genes display a wide range of expression dynamics across evolutionary time reflective of the diverse selection pressures acting on genes within primate brain tissue. Using our samples that represent a 190-fold difference in primate brain size, we identified genes with variation in expression most correlated with brain size. Our study extensively broadens the phylogenetic context of what is known about the molecular evolution of the brain across primates and identifies novel candidate genes for the study of genetic regulation of brain evolution.

## Editor's evaluation

This is an important study that represents a significant contribution to our understanding of how gene expression in the primate brain has evolved across the extant primate phylogeny. It provides solid evidence for potential links between gene expression variation and brain size, although these are somewhat limited by the focus only on adult brains, since many key changes likely occur during development. Nevertheless, both the taxonomically broad data set and the analysis are likely to

be of broad interest to the evolutionary biology, anthropology, and comparative neuroscience communities.

## Introduction

Primates are distinguished from other mammals by their large brains relative to body size (*Boddy et al., 2012*; *Martin, 1981*; *Smaers et al., 2021*). Among the diversity of primate species, there is remarkable variation in behavioral specializations, including differences in social structure, spatial, dietary and visual ecology, and locomotion (*Powell et al., 2017*; *Barton, 2012*; *DeCasien and Higham, 2019*). Despite the impressive array of cognitive attributes displayed by primates, molecular and cellular studies investigating various aspects of brain evolution tend to sample from a small number of species to address questions of how humans are unique. In large part, the emphasis on human brain evolution is warranted. Humans are unmatched in possessing exceptionally large brains and unparalleled cognitive abilities, such as language (*Konopka and Roberts, 2016*; *Rilling, 2014*). While valuable, the limited number of species included in prior research lacks a comprehensive perspective of the phylogenetic context in which the human brain evolved within the diversity of primates.

Although researchers have used a variety of approaches to assess whether the human brain is unique (*Stout and Hecht, 2017*) and how it might have evolved (*Hrvoj-Mihic et al., 2013*; *Sousa et al., 2017a*), the full potential for using gene expression to evaluate patterns of brain evolution in primates has not yet been met. Upon observing the remarkable similarity between human and chimpanzee protein sequences, *King and Wilson, 1975* proposed that the basis of the physical and behavioral phenotypic differences between these two species must be found in changes within gene regulatory regions that drive expression. Previous studies have explored how changes in regulatory regions can influence gene expression but have often sampled various organs from species across broad spans of evolutionary time, such as mammals or vertebrates (*Brawand et al., 2011*; *Breschi et al., 2016*). In studies focusing on gene expression in primate brain tissues, research has mostly focused on the neocortex and cerebellum in human, chimpanzee, and rhesus macaque (*Babbitt et al., 2010*; *Blekhman et al., 2010*; *Khaitovich et al., 2005*; *Khaitovich et al., 2004*; *Khrameeva et al., 2020*; *Konopka et al., 2012*; *Ma et al., 2022*; *Somel et al., 2009*; *Sousa et al., 2017b*). However, new insights can be gained by sampling at greater neuroanatomical resolution from a broader array of primates. Examining gene expression of the brain from a more comprehensive landscape empowers novel inquiry in primate brain evolution, including questions pertaining to the sources of variation that drive expression differences, rates of expression change across the primate phylogenetic tree, and genes that correlate with brain size across primates.

In the current study, we sampled prefrontal cortex (PFC), primary visual cortex (V1), hippocampus (HIP), and lateral cerebellum (CBL) from 18 primate species, the broadest diversity of primates sampled in any study of gene expression in the brain to date, including species from several rarely-studied lineages. Our dataset represents 70–90 million years (*Perelman et al., 2011*) of primate evolution, providing a more thorough understanding of the evolution of gene expression across primates and allowing for an unprecedented view of how gene expression in the brain has changed over time across all major clades of primate phylogeny.

## Results

### Most variation can be explained at the species level, not by brain region

To understand how gene expression in the brain has evolved across the primate lineage, we generated and analyzed RNA-seq data from 18 primate species (including five hominoids, four cercopithecoids, four platyrrhines, and five strepsirrhines, with 1–3 biological replicates) across four brain regions, including PFC, V1, HIP, and CBL (*Figure 1*, *Supplementary file 1*). The transcriptomes and gene models were assembled de novo (*Haas et al., 2013*) (see Materials and methods). We quantified the expression of 15,017 orthologs within hominoids, and 3432 on-to-one orthologs across all 18 species (*Supplementary files 2 and 3*). Variation in interspecific mammalian gene expression has been shown to be less pronounced than that observed across samples from different organs,

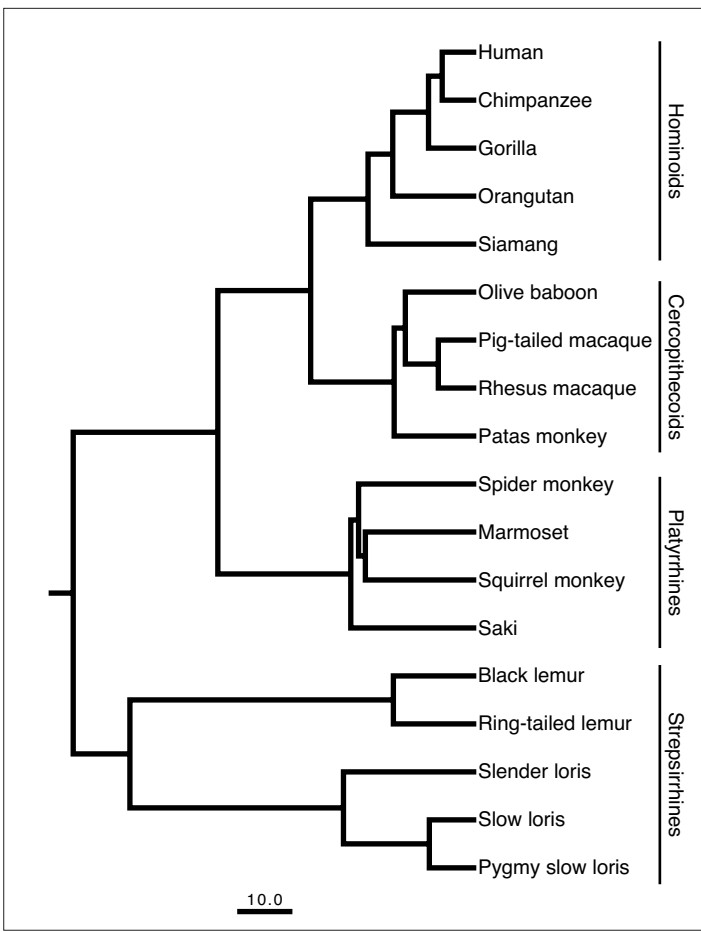

**Figure 1.** Primate phylogeny showing the eighteen species sampled in this study. The scale bar for the branch lengths represents 10 million years of evolution. The phylogenetic tree is a consensus tree of 1000 iterations produced from *10kTrees* v.3 (https://10ktrees.nunn-lab.org) based on data from GenBank. The insets demonstrate the approximate locations of the four brain regions sampled on a coronal section, midsagittal view, and lateral view (displayed left to right, respectively) of a schematized adult human brain.

reflecting the diversity of underlying organ physiology (*Brawand et al., 2011*; *Sudmant et al., 2015*). Furthermore, it has been reported that the rate of gene expression divergence evolved more slowly within the cerebral cortex and cerebellum compared to other organ systems from developmentally distinct germ layers (*Brawand et al., 2011*; *Khaitovich et al., 2005*). To explore the variability of gene expression from distinct regions of the brain across our broad sampling of primates, we constructed a pairwise distance matrix of the 500 most variable protein-coding genes based on the standard deviation of expression across samples (Methods). This subset of genes was enriched with glycoproteins, signal peptides, and plasma membrane proteins, with roles in immune function, molecular trafficking, and cell signaling. Using this distance matrix, we performed a principal coordinates analysis (PCoA) on data from all brain regions. Because our samples represent disparate regions of the same organ, we expected less variation to be attributed to brain regions than primate species or taxa, reflecting the similarity in physiology of brain tissues. Unsurprisingly, the variation from our complex gene expression dataset is represented across multiple axes of the PCoA (*Supplementary file 4*).

We plotted the first three axes and created polygons around data derived from samples sharing a common taxa (*Figure 2a–c*) or region (*Figure 2d–f*). As predicted, taxon assignment explains a large amount of variation to the dataset, with clear trends emerging independent of brain region. We find the greatest divergence in expression patterns among hominoid and strepsirrhine species, while there is more similarity observed among cercopithecoids and platyrrhines. The hominoids, displaying the greatest level of diversity of any primate phylogenetic group, demonstrate variation that is particularly apparent along Axis 1 and largely driven by human and chimpanzee expression patterns (*Figure 2A,*

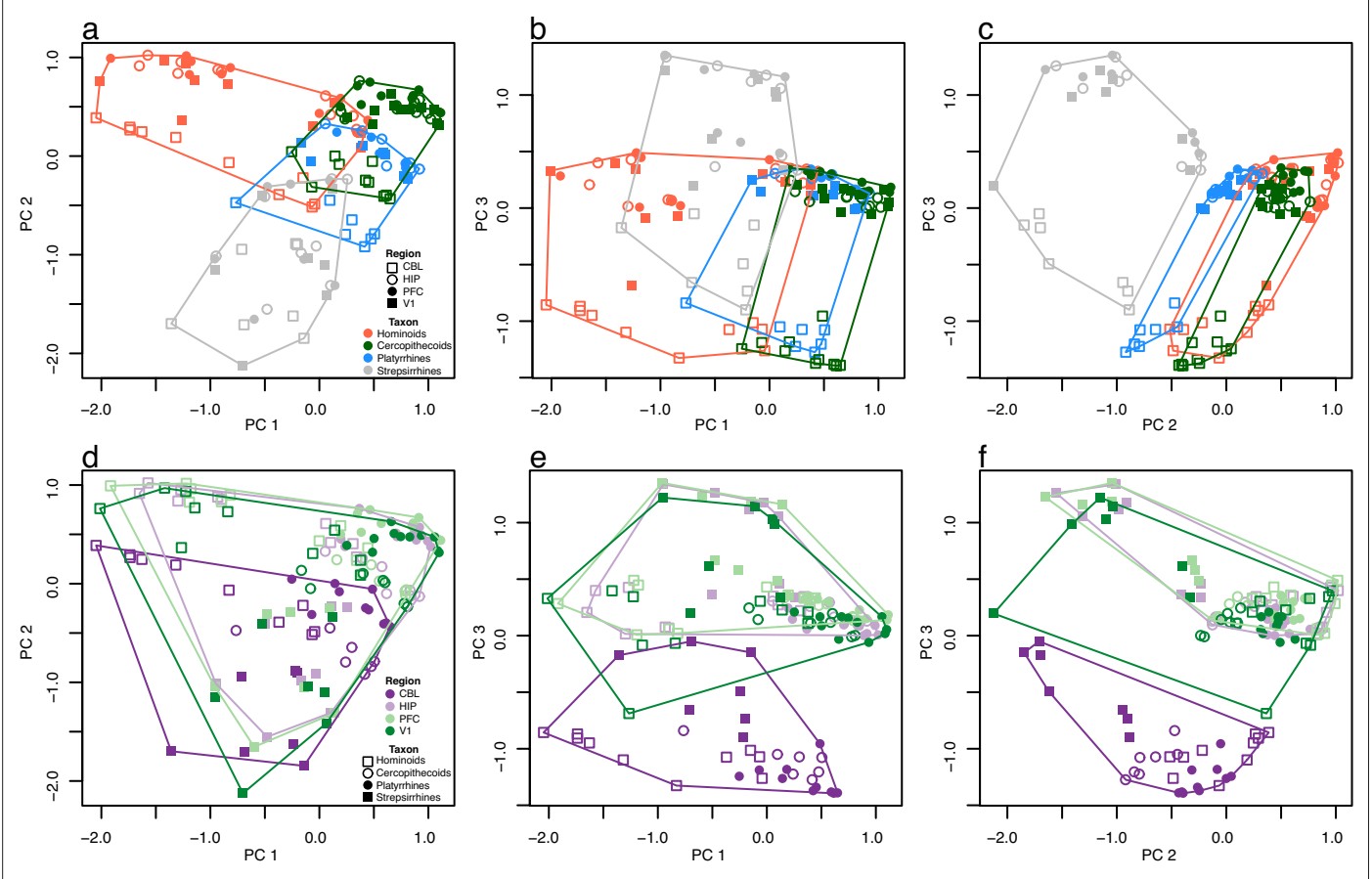

**Figure 2.** Patterns of brain gene expression across primates. The first three axes of a principal coordinates analysis (PCoA) are plotted in both rows but have different symbols and colors to emphasize expression patterns specific to taxa (upper row, **a–c**) and regions (lower row, **d-f**). Polygons in each plot surround the data points for taxa (upper row) and regions (lower row). Axes 1, 2, and 3 represent 12.8, 10.3, and 9.4% of variance, respectively.

The online version of this article includes the following figure supplement(s) for figure 2:

**Figure supplement 1.** The first three axes of the principal coordinates analysis (PCoA) are plotted in three bivariate plots.

**Figure supplement 2.** Gene expression phenogram of all sampled data.

**Figure supplement 3.** Gene expression phenograms by region.

**B**). Strepsirrhines also exhibit a large amount of variation, especially apparent along Axis 2, which can mostly be attributed to the three species of lorises. When Axis 1 and 2 of the PCoA are plotted on the same bivariate plot (*Figure 2a*), the hominoids display more variation than the strepsirrhines by about 24% (*Supplementary file 5*). However, a large portion of the variation in the strepsirrhines is attributed to evolutionary divergence over about 63 million years (since the last common ancestor of lemurs and lorises), whereas the variation within hominoids has largely accrued over only 9 million years (since humans and chimpanzees shared an ancestor with gorillas). The hominoid and strepsirrhine samples represent similar variation in terms of sex and life stage, suggesting that these factors do not account for the variability seen in these taxa. Therefore, a remarkable finding of this analysis is how much variation is represented by hominoids, despite the fact that this lineage represents a much shorter evolutionary divergence time.

## The cerebellum differs most significantly from the other three sampled brain regions

We observed trends in gene expression by brain region that are predominantly seen along Axis 3 (*Figure 2e–f*). Here, we observe that the variation attributed to CBL is distinguished from that of PFC, V1, and HIP, which are very similar in their distributions. The fact that CBL differs in its pattern of gene

expression from other brain regions (*Hawrylycz et al., 2015*; *Hawrylycz et al., 2012*; *Itō, 2012*) is not surprising given that it is the only sampled region that develops from a different part of the embryonic neural tube (namely, the hindbrain vs. forebrain) and exhibits a neuronal packing density (predominantly glutamatergic granule cells) that far exceeds these other brain regions (*Azevedo et al., 2009*; *Herculano-Houzel, 2011*; Itō, 2012). Enrichments for cerebellar gene expression reveal expression changes to categories such as 'Cell Surface Receptor Signaling Pathway,' 'Cell Projection Organization,' and 'Wnt Signaling Pathway.' This is true for humans in relation to chimpanzees as well as other species with deeper evolutionary relationships. (*Supplementary files 6 and 7*). Notably, genes involved in cell migration and cell surface receptor signaling *Emera et al., 2016* have been shown to mediate the cell-cell interactions necessary for axon guidance (*Koropouli and Kolodkin, 2014*). Although all the brain regions surveyed show some enrichment for categories related to cell signaling, the cerebellum shows a unique increase in signaling activity-related terms (both in the number of enrichment categories and the degree of change associated with these categories). This is indicative of region-specific differential expression.

## Gene expression profiles accurately replicate phylogenetic relationships

To determine whether gene expression profiles can reconstruct known phylogenetic relationships among primates, we built expression phenograms (Methods) for all brain regions combined (*Figure 2—figure supplement 2*) and each region separately (*Figure 2—figure supplement 3*). Samples that were derived from individuals of the same species tended to be grouped together, regardless of brain region, revealing that inter-individual differences are minor compared to other sources of variation. Gene expression profiles also replicated the phylogenetic relationships of closely related species (e.g. humans and chimpanzees; pig-tailed and rhesus macaques) when all regions were considered, but these relationships became less phylogenetically structured in the phenograms constructed using expression data from individual brain regions. All neighbor-joining phenograms accurately represent cercopithecoids and strepsirrhines as monophyletic groups; however, expression data produces paraphyletic groups of hominoid and platyrrhine species. This result potentially reflects the fact that taxa with longer periods of independent evolution (i.e. strepsirrhines) are more likely to show divergent patterns of gene expression than more closely related groups. Meanwhile, a more dense sampling of cercopithecoids (three individuals per species), permits a fairly accurate reconstruction of this taxon.

## Differential expression in the context of the Ornstein-Uhlenbeck model

Previous studies have used a variety of different approaches to model gene expression changes over time (*Brawand et al., 2011*; *Perry et al., 2012*). Here, we used a recently described Ornstein–Uhlenbeck (OU) model to analyze neutral and conserved processes as determined by changing gene expression levels (*Chen et al., 2019*). OU processes have been proposed to model gene expression evolution as they model both drift and stabilizing selection (*Rohlfs et al., 2014*). Previous studies have shown how models that incorporate stabilizing selection are more accurately able to predict gene expression evolution in mammals than models that account only for neutral drift (*Chen et al., 2019*). Across all expressed one-to-one orthologs represented in the sampled primates, we found that ~15–20% of genes show differential expression across all species-to-species comparisons (q-value <0.05). As expected, the relative amount of differential expression increases over evolutionary time, both between species and clades (*Figure 3* and *Figure 3—figure supplement 1* ). However, both the species and clade-wise comparisons show larger numbers of differentially expressed (DE) genes in the comparisons when strepsirrhines are included in the contrast, a result of the more than 60 million years of independent evolution of this taxon. In the prefrontal cortex, we see slightly smaller DE in humans as compared to siamang and baboon; however, in total number, this DE is not appreciably different (DE Human-Siamang n=251, DE Human-Baboon n=255) to other comparisons (DE Human-Rhesus Macaque n=315). Upset plots allow us to further analyze these unique patterns of differential expression (*Figure 3—figure supplement 2*). In the cerebellum, the lemur, baboon, and human samples are particularly unique and show relatively higher levels of DE compared to other species. However, in the hippocampus, visual cortex, and prefrontal cortex, the chimpanzees appear to show higher DE than the human samples. This suggests again that the cerebellum is a particularly unique brain structure

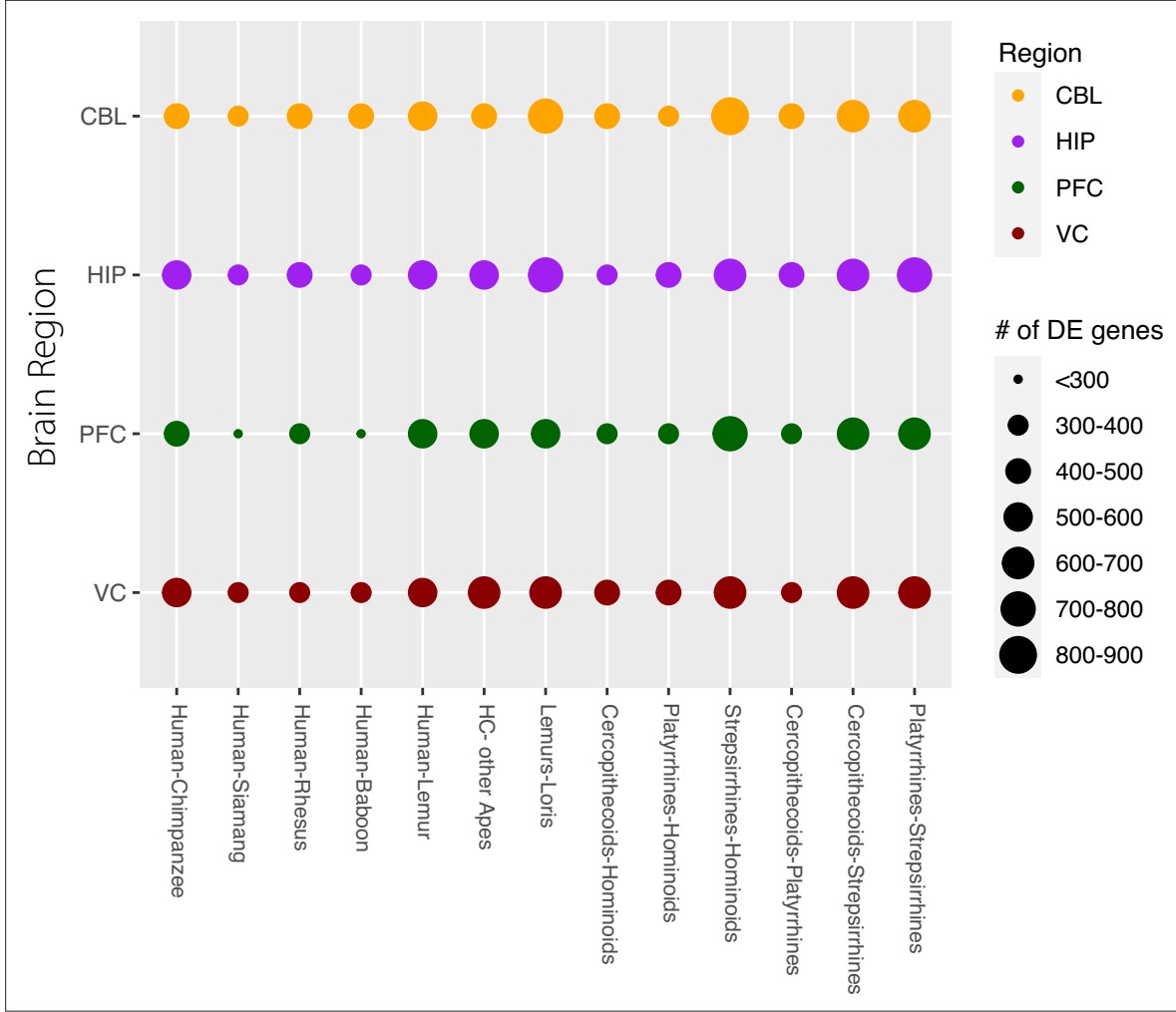

**Figure 3.** Gene counts of differentially expressed (DE) genes between species and clades. Each row represents one of the four brain regions examined. The size of the circle represents the number of DE genes seen at q<0.05 (5% FDR). The comparisons on the left are between exemplar species or sets of species, comparisons on the right are between clades of primates.

The online version of this article includes the following figure supplement(s) for figure 3:

**Figure supplement 1.** Box Plots of Pearson Rank-based Correlation Coefficients for multiple species and brain region comparisons.

**Figure supplement 2.** Upset Plot of each brain region (color) showing the shared DEGs for select species across the phylogeny.

and that chimpanzee gene expression is significantly different in the other three brain regions studied, warranting further analysis.

## Human-specific enrichment for metabolic processes, neural development, and gene regulation

When comparing gene expression in the PFC of humans relative to other primates, human PFC shows an enrichment of metabolic processes, including 'regulation of cellular metabolic process' and 'regulation of macromolecule metabolic process' (*Supplementary files 6 and 7*). Comparing human and chimpanzee PFC reveals that categories that support neural growth and development (e.g. 'neuron projection morphogenesis,' 'cell morphogenesis involved in neuron differentiation'), gene regulation, and metabolic processes are enriched as differentially expressed in human PFC relative to chimpanzees.

In addition to examining human and chimpanzee data in isolation, we also analyzed human-specific changes in expression within the context of other outgroup species. Using the Ornstein-Uhlenbeck process as a model of continuous trait evolution across our 18-species primate phylogeny, we again

observe similar categories of enriched processes for the human prefrontal cortex in comparison to that of chimpanzees. These terms include 'Nervous System Development,' 'Neurogenesis,' 'Glial Cell Differentiation,' 'Neuron Projection Morphogenesis,' 'Regulation of Gene Expression,' and 'Metabolism.' To further validate our results, we also looked at the human PFC in comparison to other primate species. In analyzing differential expression between the human and siamang PFC, we note that similar trends for enrichment are also found, such as 'Neural Growth and Development' and 'Metabolic Processes,' and 'Gene Regulation.' Of interest, under the category of 'Positive Regulation of Transcription by RNA polymerase II' we find several genes that appear to be upregulated in humans compared to siamang: APP (amyloid precursor protein, related to plaque formation in Alzheimer's disease) as well as PRKN (found to be causal in Parkinson's disease) (*Funayama et al., 2023*). This supports the idea that these enrichments are human-specific, have relevance to important human neurodegenerative disease states, and are not a reflection of changes occurring within the chimpanzee lineage.

## Broader species comparisons show similar trends across evolutionary time

Beyond human and chimpanzee comparisons, we also note many interesting broader temporal trends observed from the EVEE-based differential expression analysis. When examining Hominini (humans and chimpanzees) compared to other Great Apes, terms related to 'Regulation of Metabolic Process,' 'Nervous System Development,' and 'Biosynthetic processes' are all enriched within the hippocampus. Meanwhile, 'Negative Regulation of Synaptic Transmission' was enriched in the other ape species. In comparing the PFC of the Hominoid clade to that of the Strepsirrhine clade, we found that 'Neuron Development and Differentiation' was enriched. Overall, among various species and clade comparisons, there is a general trend of decreasing specificity in enrichment categories over increasing evolutionary time. Using PFC data, we found that the relationship between humans and chimpanzees, some of the closest relatives in our dataset, shows terms related to 'Synapse Assembly,' 'Regulation of Glial

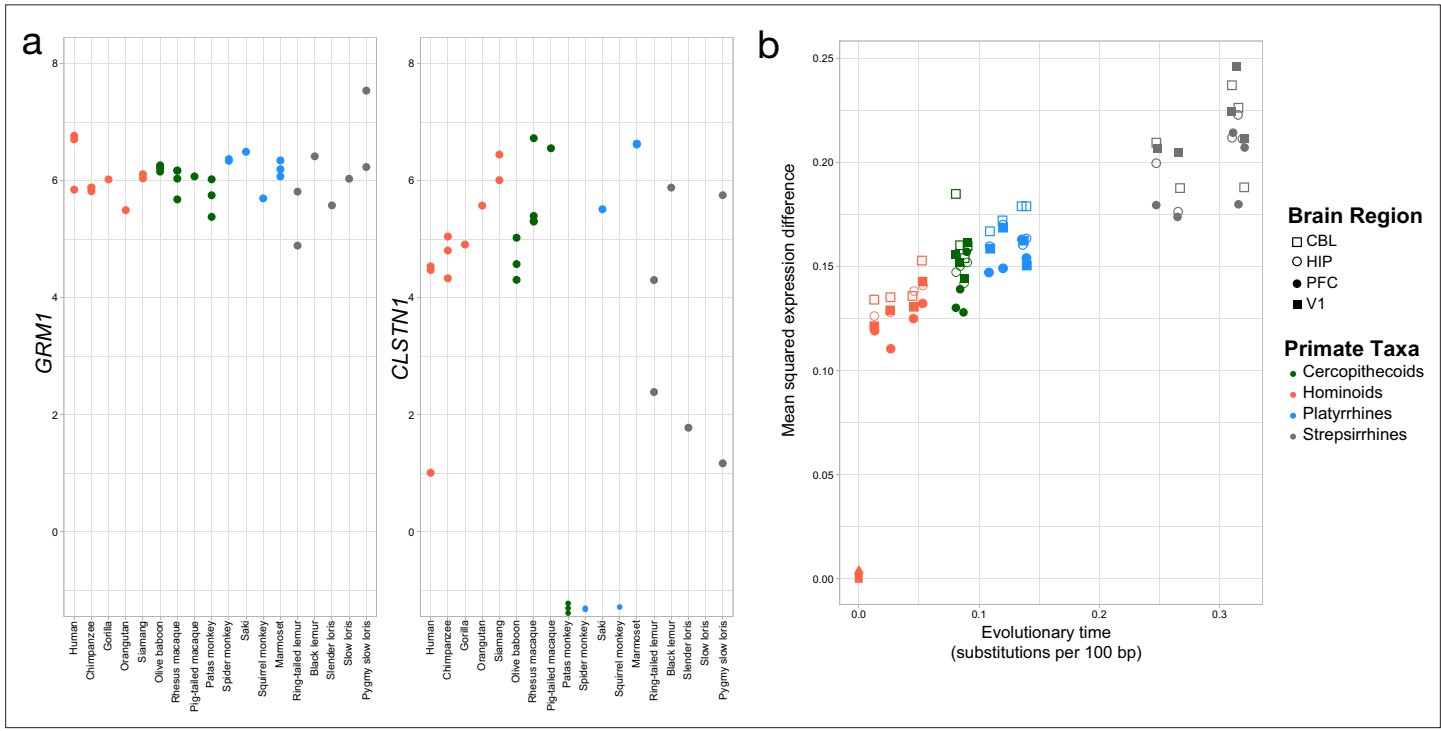

**Figure 4.** Rates of change over genes and evolutionary time. a. Exemplar genes that show constraint (left panel) and variation (right panel) across primates (colors as in *Figure 2*). b. Mean squared expression difference plotted by evolutionary distance to humans across all orthologs that were expressed. Shapes denote the four brain regions, and the colors represent the four major primate clades represented in our samples.

The online version of this article includes the following figure supplement(s) for figure 4:

**Figure supplement 1.** Region-specific plots of mean square expression differences over evolutionary time for each of the four brain regions analyzed.

Cell Differentiation,' 'Regulation of Astrocyte Differentiation,' 'Axonogenesis,' and 'Neuron Projection Morphogenesis.' Looking at a more distantly related species pair, the human and rhesus macaque comparison shows enrichment for terms related to 'Cell Growth,' 'Cell Development,' 'Biological Regulation,' 'Neuron Projection Development,' 'Regulation of Neurogenesis,' and 'Positive Regulation of RNA Biosynthetic Processes.' The most distantly related species, humans and lemurs, have overall the largest number of differentially expressed genes, and with that, the broadest categories of enrichment. These include categories such as 'Regulation of Developmental Processes,' 'Regulation of Nervous System Development,' 'Cell Development,' and 'Multicellular Organism Development'.

Our PCoA analyses showed that gene expression in brain regions sampled from the lorises (i.e. the slender loris, slow loris, and pygmy slow loris) diverged from other strepsirrhines, and other primates more generally (*Figure 2—figure supplement 1*). When strepsirrhines are compared to other primates in differential expression analyses, transcription factors, and other genes involved in gene transcription and translation and multiple biosynthetic pathways involved in cellular metabolism are among the categories of DE genes (*Supplementary files 6 and 7*).

## Evolutionary rates of expression change across clades and brain regions

Using the OU model, we found that individual genes exhibit wide variation in expression dynamics across the primate lineage (*Figure 4a*). Enrichments for genes showing low variation or stabilizing selection (q=0.05) reveal categories related to transport and cellular localization (GO Biological Processes, *Supplementary files 6 and 7*). In contrast, genes that are less constrained or neutrally evolving (q>0.05) have a number of processes related to neuron morphogenesis, plasticity, and cell death. Yet, unlike sequence evolution, gene expression is not linear across evolutionary time but a saturation point in pairwise comparisons of gene expression is reached due to stabilizing selection pressures. Here, we find that pairwise expression differences between humans and the other species increasingly diverge with evolutionary distance in all brain regions sampled (*Figure 4b*); however, these pairwise comparisons do not seem to saturate with evolutionary time across the primate comparisons (*Chen et al., 2019*). We note that the saturation of pairwise expression differences from humans may be found at a phylogenetic node ancestral to primates (*Figure 3—figure supplement 1*).

## Expression of a majority of genes evolves under stabilizing selection

We utilized EVEE-tools, developed in the context of the Ornstein-Uhlenbeck process of continuous trait evolution, to classify genes as primarily under the effects of stabilizing selection vs. a model of neutral drift (*Chen et al., 2019*). In this analysis, we found that across tissues, on average, 64% of genes fit better under stabilizing selection (64% CBL, 59% HIP, 72% PFC, and 60% V1; FDR-corrected q-value calculated via the BH procedure to correct for multiple hypothesis testing; FDR threshold of 5% to determine significance). In the context of the OU model of continuous trait evolution, we found that the overall phylogenetic signal in brain expression divergence was slightly smaller than observed with edgeR over the entire combined dataset (EVEE: 4–6.7% across all single species comparisons using a logFC of +/-2; edgeR: 15–20%). However, the relative amount of differential expression does increase gradually with evolutionary distance (as expected based on results in edgeR). The only exception to this is the human and chimpanzee comparison, which shows considerable variation in differential expression across the four brain regions. Representing such a short period of evolutionary divergence, this increase in DE suggests direction selection within that lineage.

## Evidence of potential directional selection in human and chimpanzee data

To better understand the outlier gene expression in this dataset, and overall to gain insight into genes that may be subject to directional selection pressures, we again used the EVEE-tools OU-based model to score our dataset for outlier expression. This requires the determination of the evolutionary mean and variance for each gene across our entire expression dataset, from which we then can compare to individual species expression data. Using this method, we found a small subset of genes to be defined as having an expression that deviates from the optimal OU-distribution (Z-score >2 or < –2, p-value <0.05) (*Figure 4Figure 4—figure supplement 1*). It is important to note that a significant FDR (<5%) was not reached in this dataset, among all species comparisons. This is expected based on previous

applications of this package, in which a mammalian dataset was also unable to reach an FDR below 18% (*Chen et al., 2019*). This is likely a reflection of the limitations of our phylogeny, and suggests that future projects should aim to sample even more broadly.

In our outlier expression analysis, we found that patterns of outlier gene expression occur in a species and tissue-specific manner (*Supplementary file 10*). For example, in the PFC, the chimpanzee and marmoset samples appear to have the highest number of outlier genes deviating from average expression patterns (compared to human, siamang, baboon, rhesus macaque, and marmoset samples). We found this to be true regardless of how the dataset was normalized in order to determine the average expression for each gene (via defining a reference species). This is particularly interesting and, in combination with differential expression analysis, highlights the chimpanzee PFC as a particularly divergent structure. In contrast to the PFC, outlier analysis in the CBL reveals that the human and lemur samples have the highest number of outlier genes.

Upon gene set category enrichment analysis, we find that many of these genes that are deemed 'outliers' are related to functions in development, transcription, nervous system development, neurogenesis, and metabolism. For example, the chimpanzee PFC shows a significant upregulation in genes involved in energy storage and transfer, such as ETFA and NDUFS4, which are both involved in electron transport for ATP generation, as well as ANKH, implicated in phosphate transport (*Szeri et al., 2020*; *Henriques et al., 2021*; *Shil et al., 2021*) We also see that the chimpanzee PFC shows unique expression patterns in genes related to synaptic activity and neurotransmitter release, including significant downregulation of GABRA4 and SYN1 (*Fassio et al., 2011*; *Fan et al., 2020 Fan et al., 2020*; *Fassio et al., 2011*). In contrast, the human CBL and PFC both display a significant upregulation of genes related to Amyloid protein production (APP), a major component of many neurodegenerative diseases with functions in synaptic signaling (*O'Brien and Wong, 2011*). Unique to the human CBL we also see enrichments in genes related to neurogenesis and synaptic activity, including SDK1, FZD5, and CDH10 (*Redies et al., 2012*; *Slater et al., 2013*; *Bagot et al., 2016*) This is suggestive of directional selection pressures occurring in a tissue-specific manner and encourages future investigation of these outlier genes.

## Implications of using humans as a reference species

We continue to use humans as a reference species in these analyses as, compared with other primates, humans have exceptionally large brain sizes and unique cognitive abilities. However, we do recognize that there are some implications for having humans as a reference, especially given our data that would suggest human gene expression as being largely different from the rest of our primate dataset. To address this, we repeated analyses using EVEE-tools by including two additional reference species: siamang and rhesus macaque. The percentage of genes that fit better under the model of stabilizing selection (in comparison to neutral drift) is not statistically different from those observed when using humans as a reference (on average, 58–64% across all four major brain regions). We additionally looked at pairwise species comparisons to determine if the general trends of directional selection and differential gene expression were comparable to the human-reference data, and again confirmed that the effects of the reference species used for normalization here are negligible in terms of analyzing differential expression across our primate tree.

## Correlation of gene expression to brain size and their change over evolutionary time

Because an increase in absolute brain size is one of the most striking characteristics of humans, we asked what subset of genes is correlated with this important biological trait across the primate tree. Our dataset provides a unique opportunity to evaluate this question, since the average brain size across our study species varies by ~190 fold. Using multivariate analysis (Methods), we defined gene lists most strongly correlated with brain size in each brain region (*Supplementary file 8*). Results indicate that the same genes rank among those with the strongest positive correlation in PFC, V1, and HIP. CBL also shares some of these same genes but includes more variation among the genes most strongly correlated to overall brain size than the other three brain regions (*Figure 5*, *Supplementary file 9*), potentially reflecting the more recent expansion of the CBL relative to the rest of the neocortex (*Miller et al., 2019*; *Smaers et al., 2018*).

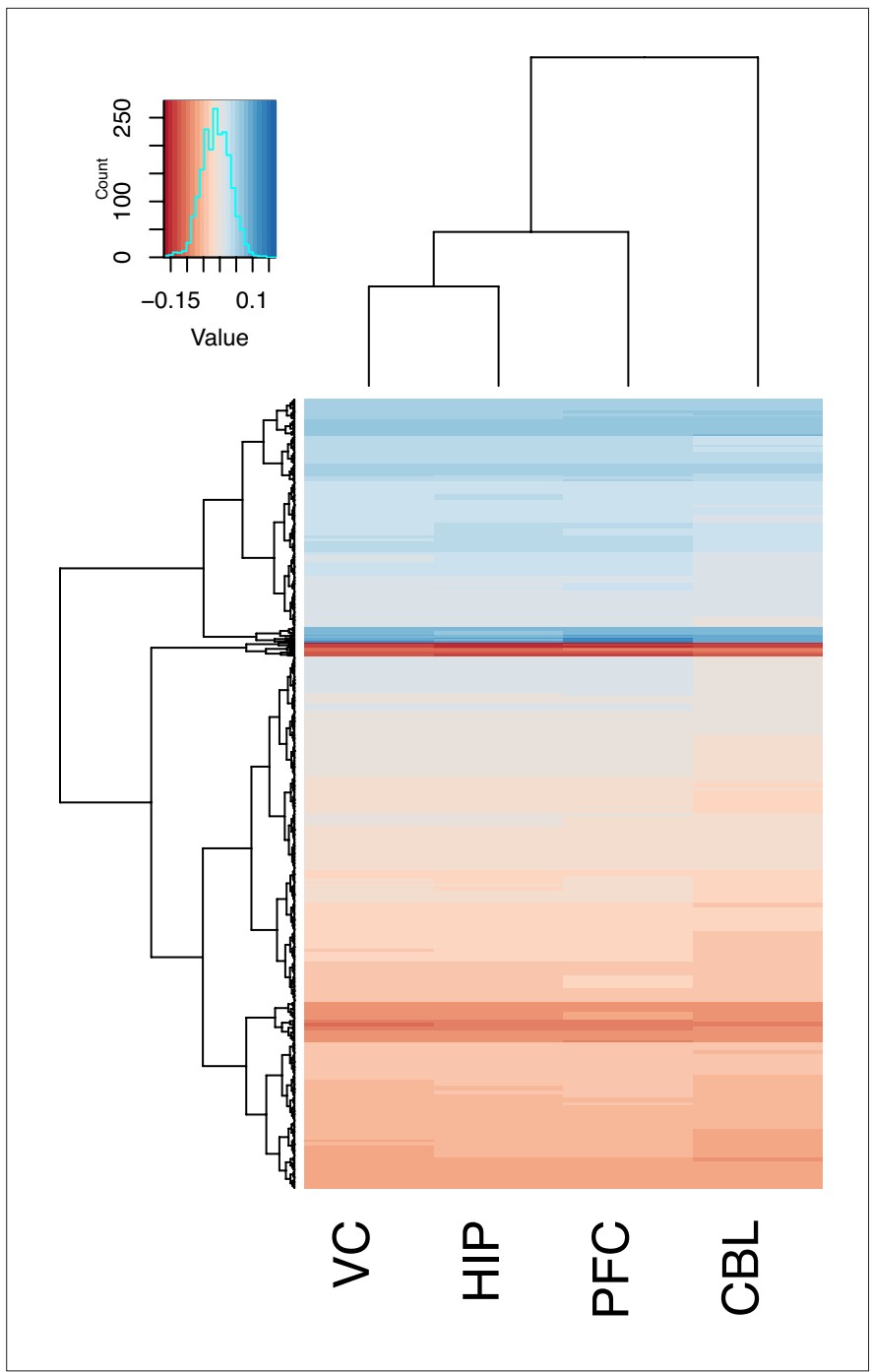

**Figure 5.** Gene correlated with brain size by region. A clustering and heatmap of the loadings from PC2 of genes for the four regions examined (V1, HIP, PFC, and CBL).

The online version of this article includes the following figure supplement(s) for figure 5:

**Figure supplement 1.** principal coordinates analysis (PCoA) of expression data from human samples of all four brain regions and primary neurons and astrocytes.

## Discussion

We performed RNA-seq on four brain regions from 18 primate species, representing the broadest sampling for any gene expression study in primate brain tissue to date. Through more representative sampling of primate species, we found substantial variation in gene expression levels within the

hominoid and strepsirrhine lineages, with the diversity among hominoids particularly impressive due to the recent divergence of this taxon. Using the OU model, we found that a substantial proportion of genes showed differential expression across species. The relative amount of differential expression increased over evolutionary time, both between species and clades. Additionally, when comparing gene expression across broader species and clade comparisons, we observed trends related to brain development, nervous system regulation, and cellular metabolism.

Our findings point to human-specific enrichment for metabolic processes, neural development, and gene regulation. The considerable diversity of gene expression in human and chimpanzee brain tissue has profound implications for understanding the distinct evolutionary processes that have acted upon the brains of the ancestral species of these two lineages. We observed a wide variety of expression dynamics of individual genes in the pairwise comparisons of humans to other primate species. Enrichments for genes under stabilizing selection indicated processes related to transport and cellular localization, while less constrained genes were associated with neuron morphogenesis, plasticity, and cell death. Importantly, gene expression evolution did follow a linear pattern, but did not reach a saturation point due to stabilizing selection pressures as seen in other studies (*Chen et al., 2019*). Less constrained, neutrally evolving patterns appeared to be the most prevalent pattern in each brain region studied, preventing a saturation point of stabilizing pressures to be reached within Primates with increased phylogenetic distance from humans. Lastly, we identified genes that are correlated with brain size across all major primate taxa, providing candidates for further inquiry.

Our deeper analysis of gene expression has revealed evolutionary patterns that were inaccessible with a more limited sampling of primate brain tissue. We anticipate that the candidate genes and data provided by this study will serve as a resource for many other lines of inquiry into human and non-human primate brain evolution.

## Materials and methods
### Biological sample collection and RNA extraction
The sample includes brain tissue from human and nonhuman primates. All samples were obtained from adult individuals free from known neurological disease. If available, the right hemisphere was preferentially sampled. Human brain samples were obtained from the National Institute for Child Health and Human Development Brain and Tissue Bank for Developmental Disorders at the University of Maryland (Baltimore, MD). Chimpanzee brain tissue was obtained from the National Chimpanzee Brain Resource (https://www.chimpanzeebrain.org, supported by NIH grant NS092988). All other sources of brain tissue are listed in *Supplementary file 1*.

From each individual, we sampled four regions of the brain, including PFC, V1, HIP, and CBL. PFC was sampled from the frontal pole, corresponding to Brodmann's area 10 in humans. In other primates, the PFC region sampled more broadly encompassed prefrontal cortical areas but was limited to the most anteriorly projecting part of the frontal pole. All V1 samples were dissected around the calcarine sulcus to include primary visual cortex (Brodmann's area 17). Samples from both PFC and V1 contained all cortical layers and a small amount of underlying white matter (<10%). The HIP was sampled from the medial aspect of the temporal lobe and included all hippocampal subfields. The CBL was sampled from the most laterally projecting region of lateral hemisphere in all primates. In humans, the CBL region corresponded to Crus I or Crus II. CBL samples contained all layers of cerebellar cortex and a small amount of underlying white matter (<10%). Each sample was briefly homogenized using a Tissuelyzer (Qiagen), and the total RNA was isolated using an RNAeasy kit (Qiagen) with a DnaseI treatment.

### Library preparation and sequencing
Single-end RNA-seq libraries were made using the NEBNext mRNA Library Prep Reagent Set for Illumina. Libraries were prepared in batches of 4–8 samples of randomly sampled species and brain regions. Library sizes were checked on the Bioanalyzer (Agilent). RNA-seq libraries were multiplexed on the NextSeq500 (Illumina) in the Genomics Resource Laboratory at the University of Massachusetts Amherst, also randomly distributed across NextSeq500 runs. All fastq files have been submitted to the SRA: https://www.ncbi.nlm.nih.gov/bioproject/?term=PRJNA639850.

## Mapping and transcriptome analysis

Sequencing reads were assembled into species-specific transcriptomes (containing the reads from all four brain regions) using Trinity (*Grabherr et al., 2011*). With Trinity assembly, fragments of the original RNA reads for each species are compiled and clustered into groups based on sequence similarity, which eventually are extended to reconstruct full-length transcripts (*Grabherr et al., 2011*). These transcripts were then blasted against the human blast nt database, with the alignment thresholds for the top hits from different clades listed in *Supplementary file 2*, similar to the approach in *Perry et al., 2012*. Individual libraries were then mapped to the species-specific transcriptome using bowtie (*Langmead et al., 2009*) in RSEM, and count tables were generated using RSEM (*Li and Dewey, 2011*). Orthology assignments were additionally checked using the Ensemble one-to-one orthology alignments as a guide for the subset of species with a publicly available genome. Transcriptome quality was assessed using BUSCO to determine assembly and annotation completeness (*Seppey et al., 2019*), *Supplementary file 3*. We recognize that some of the species transcriptomes show relatively lower BUSCO completeness scores (namely the Slender Loris at 35.8% complete). We hypothesize that this is likely a reflection of the limited tissue sampling in this dataset of only brain tissue. Previous studies have shown that transcriptome assemblies from single tissue regions on average have lower completeness scores than assemblies composed of reads from a variety of tissue types (*Simão et al., 2015*). This is likely a reflection of tissue-specific gene expression. As evidence of this, we further analyzed reads deemed as 'missing' by BUSCO and found that many of these showed little to no expression across the human brain (*Uhlén et al., 2015*) and Human Protein Atlas proteinatlas.org. We do not make any major conclusions about the Loris species in this manuscript and thus do not believe these BUSCO scores significantly affect the conclusions made in this manuscript. We only consider Loris data in concert with Lemurs, which by comparison have much more complete transcriptomes.

## Distance-based data analyses (PCoA and phenograms)

We performed principal coordinates analyses (PCoA) based on a pairwise distance matrix of all 137 samples. The distance matrix was comprised of the top 500 most variably expressed protein-coding genes. Pairwise distances were calculated by leading log2 fold change, providing a symmetrical representation of the expression ratio centered around 0 (i.e. log2(2)=1 while log2(0.5) = –1) (*Robinson et al., 2010*). Creating the distance matrix and plotting the PCoA were performed using the plotMDS.DGElist function (based in limma) in the edgeR package in R. Although variation is represented across more than 20 axes (*Supplementary file 4*), the first three axes were plotted to compare patterns across primate taxa and brain region sampled (*Figure 2*, *Figure 2—figure supplements 1–3*). Polygons overlap the data points representing taxa or brain region. The area of each polygon was computed using functions in the sp package of R. The chull() function was used to define the points around the perimeter of each polygon, and the Polygon() function calculated the area of each. Relative areas of each polygon are listed in *Supplementary file 5*. *Figure 3* displays the same data as the PCoA but uses an array of colors allowing the data from each individual species to be visualized.

The same log2 fold change distance matrix was then used to create phenograms representing the similarity of gene expression profiles among samples. The minimum distance neighbor-joining function in the 'ape' package of R created a tree based on the method proposed by *Saitou and Nei, 1987*. The boot.phylo function estimated the reliability of given nodes of the tree by resampling over 1000 iterations. Although our objective in this analysis was to investigate patterns of evolution across the primate order and the brain regions, our sample included multiple individuals from the same species. By treating these samples separately, our analyses represent both within- and between-species variability in gene expression over time.

### Analyzing differential expression

Counts were filtered and normalized using edgeR (*Robinson et al., 2010*), with any multispecies comparisons using the GLM functionality (*McCarthy et al., 2012*). Gene Ontology enrichments were performed using the DAVID gene ontology tool 6.8 (*Huang et al., 2009a*; *Huang et al., 2009b*) and g:Profiler (*Reimand et al., 2016*). *Supplementary files 6 and 7* shows results from ordered g:profiler enrichments (g:GOSt) performed on DE genes where q<0.05 (note, this is not ranked on polarity of expression, just absolute change) with all genes expressed in this study used as background.

Differential Expression was also analyzed using the package EVEE-tools to incorporate the Ornstein-Uhlenbeck model (*Chen et al., 2019*). Evolutionary means and variance values were calculated for each gene across the entire phylogeny for individual brain-region datasets as well as the entire RNA-seq bulk dataset in the context of the Ornstein-Uhlenbeck model. Differential expression was determined using a multivariate OU model. Regimes were defined by species/clades of interest (i.e. in human vs. chimpanzee comparisons, the human samples represent one regime in one model to compare to all other samples, represented by a second regime. In comparison, the chimpanzee samples would be represented as an additional model and a separate regime.) p-values were calculated to represent the fit of each OU model (i.e. Human or Chimp-specific expression patterns) against a Brownian motion model. These were then corrected for multiple hypothesis testing using the Benjamini-Hochberg FDR procedure. We used an FDR threshold of 0.05 to define significance. Additionally, Akaike and Bayesian Information Criterion (AIC and BIC) scores were calculated for each gene for each model, and only genes with AIC and BIC scores significant against the null were considered in further analysis. Directionality of differential expression was further determined by comparing estimated mean expression levels for each regime in each model.

We also looked at differential expression beyond pairwise comparisons, again using EVEE-tools, to see how each individual species differed across the entire dataset. With this, one species of interest was treated as a single regime while all 17 other species were grouped as a second regime. The same criteria as above were used to determine significant differential expression, and this data was utilized to construct UpSet plots using the UpsetR package (*Conway et al., 2017*). We included only relevant species in these UpSet plots (representing each major primate clade) to simplify the graphs. Each graph is also separated by brain region, similar to previous analyses (*Figure 3—figure supplement 2*).

We also analyzed sources of variation in our data to determine how significant of an effect species differences have compared to other factors, including primate families and individual variation. For this, we conducted a correlation analysis utilizing the non-parametric Mann-Whitney U (MWU) test on the differentially expressed genes between humans and chimpanzees to other primate species. We focused this analysis on only those species with three individuals per brain region (olive baboon, rhesus macaque, and lemurs). Using the set of human-chimpanzee DEGs, we calculated the Spearman rank-based correlation coefficient between each species to either human or chimpanzee expression. We determined whether or not these correlation coefficients were significantly different across species and across brain regions using the Mann-Whitney U test. We determined that there was no significant correlation between any of the three species to human or chimpanzee expression in any of the four brain regions (*Figure 3—figure supplement 1*; *Yapar et al., 2021*). This suggests that the expression profiles of these more distantly related primates are equally similar to human and chimpanzee expression patterns. We also conducted additional Analysis of Covariates (ANCOVA) to confirm that other factors, namely age and sex, were not significant sources of variation in our gene expression analyses. These analyses, along with our PCoA plots, show that taxon identity and brain region are the two most significant determinants for DE. Additionally, differences in samples from the same individual are defined by differences at the level of brain region. ANCOVA analyses showed minor residual effects that we deemed as random and were not further analyzed for the purposes of this manuscript.

Outlier expression was determined using the scoreGenes.R script from the EVEE-Tools script suite (*Chen et al., 2019*). We compared the total dataset-normalized mean expression of a single gene to that of a single select species, in the context of the overall mean expression and variance across the dataset. We specifically analyzed outlier expression in a brain region-specific manner, looking at the datasets subset by individual brain region. To normalize the expression of the entire dataset, a single species was selected to be used as a reference in TMM normalization. For all outlier analyses except for humans, the human samples were used as a reference. For the human outlier analysis, the rhesus samples were used as a reference for normalization. Importantly, we tested the use of different reference species for dataset normalization and did not find a significant difference in the number of outlier genes and the enrichment categories associated. Z-scores were calculated for genes whose expression patterns fit an OU model (in comparison to the null model of Brownian Motion) and whose evolutionary mean is above 5 CPM for the entire dataset (*Chen et al., 2019*). In order to determine significant outliers, an FDR threshold of 0.05 was again employed, however at this level limited significance was found. *Supplementary file 10* shows the results of this analysis for the CBL and PFC brain regions.

## Phylogenetic and evolutionary distance analysis

Categorical enrichments for the contrasts between species and clades are in *Supplementary file 6*. The phylogenetic tree for primates was downloaded from the UCSC Genome Browser (30 primate species) (*Kent et al., 2002*). Distances between species were extracted using the Environment for Tree Exploration Toolkit (*Huerta-Cepas et al., 2010*). The residuals and mean squared expression differences of all orthologs across 18 species were found using the package EVEE (*Chen et al., 2019*), and in all contrasts, humans were used as the reference species. We then analyzed the subset of genes showing either broadly defined conserved (low) or neutral (higher) variation across species (low <q = 0.05, high q>0.05), with categorical enrichments for these two groups in *Supplementary file 7*.

## Comparisons of the heterogenous tissues used and single-cell gene expression data

In any study that derives results from homogenized tissue samples, the composition heterogeneity of the samples may drive differences in gene expression (*Montgomery and Mank, 2016*). To address this issue, we compared the expression of our tissue samples to recent studies that have performed single-cell RNA-seq on neurons and astrocytes. RNA-seq data from primary neurons and astrocytes were obtained from NCBI's Gene Expression Omnibus (GEO) and processed in the same manner as the tissue samples for all human samples. These included four hippocampal astrocytes, four cortical astrocytes, and one cortical neuron from *Zhang et al., 2016* (GEO accession number GSE73721) and three pyramidal neuron samples isolated from an unspecified brain region by the ENCODE project (*ENCODE Project Consortium, 2012*; *Davis et al., 2018*) (accession numbers GSM2071331, GSM2071332, and GSM2071418). Only genes with counts greater than zero in all samples and (CPM)>1 in all 23 samples were included in this analysis (n=7111). A PCoA was made from a distance matrix of the top 500 most variably expressed genes by the pairwise biological coefficient of variation (method = "bcv") across samples (*Robinson et al., 2010*). Creating the distance matrix and plotting the PCoA were performed using the plotMDS.DGElistfunction in the edgeR package in R. The PCoA of our human samples in comparison to primary neurons and astrocytes suggests that our heterogeneous tissue samples are not biased to contain more neurons or astrocytes as compared to each other (i.e. one tissue is not biased within this small sample set) (*Figure 5—figure supplement 1*), and is consistent with other neural cell and brain tissue comparisons (*Khrameeva et al., 2020*).

Additionally, prior knowledge of variation across primates in cell type composition of the brain is informative in interpreting bulk RNA-seq data. It is well known that neuron densities tend to decrease as brain size increases (*Sherwood et al., 2020*). This suggests that larger brains accommodate a smaller number of neurons per unit volume compared to smaller brains. However, it is interesting to note that other structural elements, such as astrocytes (*Munger et al., 2022*), microglia (*Dos Santos et al., 2020*), and synapses (*Sherwood et al., 2020*), exhibit a relatively invariant density per unit volume across species. Despite changes in brain size, these essential components involved in neural communication and support maintain a consistent presence, emphasizing their crucial role in brain function regardless of the species' brain size. Thus, based on this previous research, in cross-species comparisons of bulk tissue, we can tentatively interpret differences in gene expression to reflect generally similar proportions of major cell types, except for neurons, which are expected to decrease in proportion with larger brain sizes.

## Brain size analyses

Average species endocranial volumes (ECVs) were obtained primarily from *Kamilar and Cooper, 2013* and *Isler et al., 2008* from the mean of male and female volumes. ECVs were used since reliable brain size data does not exist for all species samples. The data for human ECV (also averaged from male and female data points) was previously published by *Coqueugniot and Hublin, 2012*. *Isler et al., 2008* reported a minimal error when ECV was transformed to brain mass using the correction factor of 1.036 g/ml (*Stephan, 1960*), and we used this conversion to obtain brain mass estimates from ECVs for all species (*Supplementary file 8*).

Within each brain region, we performed a PCA on the species average gene expression of the 500 most variable genes by standard deviation, using the prcomp function in R. For each regional PCA, 14 PCs were required to account for about 90% of the variance in gene expression. We performed multiple regression analyses to determine which of the PCs could predict brain size. Using all 14 PCs

accounted for at least 95% of the variation in each brain region. Akaike information criterion was applied. However, it was noted that for each brain region, PC2 was the most predictive of brain size by low (regional adjusted $R^2$ values for PC2 against brain size were: PFC, 0.42; V1, 0.56; HIP, 0.50; CBL, 0.36). The 500 genes and their loadings on PC2 are listed in *Supplementary file 9*. Across the four sampled regions, we find general uniformity in the extent to which individual genes affect brain size, but the cerebellum displays the most unique signature of the regions sampled (*Figure 5*).

## Acknowledgements

We would also like to acknowledge our funding from NSF BCS-1750377, the Wenner Gren Foundation, James S McDonnell Foundation (220020293), NSF INSPIRE (SMA-1542848), NIH (NS-092988), and a fellowship to KR from NIH T32 GM135096. https://doi.org/10.37717/220020293.

## Additional information

### Funding

| Funder | Grant reference number | Author |
| --- | --- | --- |
| National Science Foundation | BCS-1750377 | Courtney C Babbitt |
| National Institutes of Health | T32 GM135096 | Katherine Rickelton |
| James S. McDonnell Foundation | https://doi.org/10.37717/220020293 | Chet C Sherwood |
| National Institutes of Health | NS-092988 | Chet C Sherwood |
| National Science Foundation | SMA-1542848 | Chet C Sherwood |

The funders had no role in study design, data collection and interpretation, or the decision to submit the work for publication.

### Author contributions

Katherine Rickelton, Formal analysis, Visualization, Methodology, Writing – review and editing; Trisha M Zintel, Data curation, Formal analysis, Writing – review and editing; Jason Pizzollo, Formal analysis, Investigation, Methodology; Emily Miller, Formal analysis, Visualization; John J Ely, Mary Ann Raghanti, William D Hopkins, Patrick R Hof, Resources, Writing – review and editing; Chet C Sherwood, Conceptualization, Resources, Investigation, Writing – review and editing; Amy L Bauernfeind, Conceptualization, Investigation, Writing – original draft, Writing – review and editing; Courtney C Babbitt, Conceptualization, Formal analysis, Investigation, Writing – original draft, Writing – review and editing

### Author ORCIDs

Katherine Rickelton (ID) http://orcid.org/0000-0001-6059-7505
Amy L Bauernfeind (ID) https://orcid.org/0000-0001-8518-3819
Courtney C Babbitt (ID) http://orcid.org/0000-0001-8793-4364

### Decision letter and Author response

Decision letter https://doi.org/10.7554/eLife.70276.sa1
Author response https://doi.org/10.7554/eLife.70276.sa2

## Additional files

### Supplementary files

- Supplementary file 1. Species and sample information.
- Supplementary file 2. Percent identity thresholds used for orthology assignments for each species.

- Supplementary file 3. BUSCO Scores against Mammalian lineage (mammalia_odb10).
- Supplementary file 4. Eigenvalues, percent variance, and cumulative variance across disparate axes of the principal coordinates analysis (PCoA) of the 500 most variable genes from all data sampled.
- Supplementary file 5. Area occupied by each polygon representing either taxa or brain region in the principal coordinates analysis (PCoA) of the 500 most variable genes from all data sampled.
- Supplementary file 6. Enrichments are performed on an ordered query of differentially expressed (DE) genes where q<0.05 (this is not ranked on higher or lower expression, just absolute change).
- Supplementary file 7. Enrichments are performed on an ordered query of genes comparing those with a higher degree of fit for the Ornstein–Uhlenbeck (OU) model (stabilizing expression) with those that fit the Brownian Motion model (neural expression).
- Supplementary file 8. Species endocranial volumes (ECVs) were obtained from *Isler et al., 2008* and *Coqueugniot and Hublin, 2012*. Reported volumes are averages of male and female data. ECVs are transformed to brain mass using the *Stephan, 1960* correction factor of 1.036 g/ml.
- Supplementary file 9. PC2 results from the brain size analysis by region.
- Supplementary file 10. Outlier Expression Analysis using the scoreGenes.R script in the EVEE-tools script set.
- MDAR checklist

## Data availability

Sequencing data have been deposited in the Short Read Archive: BioProject PRJNA639850.

The following dataset was generated:

| Author(s) | Year | Dataset title | Dataset URL | Database and Identifier |
|---|---|---|---|---|
| Rickelton K, Zintel TM, Pizzollo J, Miller E, Ely JJ, Raghanti MA, Hopkins WD, Hof PR, Sherwood CC, Bauernfeind AL, Babbitt CC | 2023 | Tempo and mode of gene expression evolution in the brain across primate tree | https://www.ncbi.nlm.nih.gov/bioproject/?term=PRJNA639850 | NCBI BioProject, PRJNA639850 |

The following previously published datasets were used:

| Author(s) | Year | Dataset title | Dataset URL | Database and Identifier |
|---|---|---|---|---|
| Zhang Y, Sloan SA, Clarke LE, Caneda C | 2015 | RNA-Seq of human astrocytes | https://www.ncbi.nlm.nih.gov/geo/query/acc.cgi?acc=GSE73721 | NCBI Gene Expression Omnibus, GSE73721 |
| Consortium ENCODE Project | 2016 | single cell RNA-seq from pyramidal cell (ENCLB928LID) | https://www.ncbi.nlm.nih.gov/geo/query/acc.cgi?acc=GSE78331 | NCBI Gene Expression Omnibus, GSE78331 |

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
