## [Editor Report]

This is an important study that represents a significant contribution to our understanding of how gene expression in the primate brain has evolved across the extant primate phylogeny. It provides solid evidence for potential links between gene expression variation and brain size, although these are somewhat limited by the focus only on adult brains, since many key changes likely occur during development. Nevertheless, both the taxonomically broad data set and the analysis are likely to be of broad interest to the evolutionary biology, anthropology, and comparative neuroscience communities.

---

## [Decision Letter]

**Decision letter after peer review:**

Thank you for submitting your article "Tempo and mode of gene expression evolution in the brain across Primates" for consideration by *eLife*. Your article has been reviewed by 2 peer reviewers, and the evaluation has been overseen by a Reviewing Editor and Patricia Wittkopp as the Senior Editor. The following individuals involved in review of your submission have agreed to reveal their identity: Mehmet Somel (Reviewer #2).

Essential revisions:

1) Control for phylogenetic structure in analyzing gene expression changes across the phylogeny, especially in association with brain size. Strongly consider whether phylogenetically informed analysis methods provide greater insight (and reduce the potential for confounding) over pairwise comparisons, especially given advantages afforded by broad sampling across the primate order.

2) Provide key missing detail on criteria for defining orthologues; demonstrate robustness of results to decisions about cut-offs (e.g., for defining highly variable genes) and cell type compositional heterogeneity between species.

3) Address methodological questions raised by the reviewers regarding multiple hypothesis testing correction and enrichment analyses.

4) Test human-specific changes in expression in relationship to outgroup species as well as chimpanzee.

*Reviewer #1 (Recommendations for the authors):*

1. Trinity output is extremely noisy and returns many isoforms with poor support, or virtually undistinguishable from each other except for a couple of base pairs here and there, especially when run in a de novo mode. How did the authors prioritise the many isoforms, and determine which are credible and worth analysing further? Similarly, since many of the primates lack a reference genome, how did the authors define the set of 3432 testable one-on-one isoforms, or the 15017 genes testable across hominoids? Is it simply on the basis of pairwise orthology to human using BLAST or…? Did the authors control for gene duplication (eg reciprocal BLAST)? In addition, the thresholds for orthology seem very permissive, on the basis of overall coding sequence conservation amongst mammalian species (eg see supp Figure 1B of Chen et al. 2019, where mean coding sequence identity between humans and non-primate one-to-one orthologs is ~85%).

2. Many conclusions are based on data from the top 500 most variable genes (generally defined on the basis of SD, but not always – the sanity check against scRNA from astrocytes and neurons uses CoV). Why 500? How robust are all of the results presented to this choice of threshold? Is there a particular species driving this variance (on the basis of PCoA results I dare speculate it's chimpanzees and humans)? How do these observations (PCoA, phenograms) compare to those made on the entire dataset of 3432 testable genes across the entire dataset? It seems to me that the number of genes in the whole data set is not so large as to be worth focusing only on an arbitrarily chosen threshold, and that it would be more informative to consider the entire dataset in these analyses. Similarly, when looking at positive selection, the authors only focus on the top 200 genes DE between human and chimpanzee. Why these limitations?

3. The authors test the possibility that compositional heterogenity across samples may be biasing their results (line 723 onwards), which I commend, but nonetheless find somewhat incomplete as it currently stands. The only test performed is a comparison of human bulk RNA samples to a small number of astrocyte and neuron RNA-seq samples, which they say proves their tissues are not biased towards either cell type. But combining bulk RNA and scRNA data is not trivial, and since the PCoA shows samples clustering by technology/study (I presume the outlier neuron comes from the same dataset as the astroctyes), it's unclear how to interpret it. Regardless, there's no mention of what I think is the more interesting source of heterogeneity: is there an expectation that the composition of the same tissue type would vary substantially between species? e.g., an excess of, say, glia in the PFC of humans relative to lemurs or loris that could skew results (totally hypothetical example)? I am not sure if this is possible to taste with existing datasets, but it seems to at least be worth discussing?

4. DE testing: As above, I would like to see more detail in the methods here to better contextualise the results. First, I do not think CPM is the accurate unit for comparison here, since it does not control for differences in gene length between species, which may be substantial at some of the orthology thresholds set by the authors – RPKM might be better suited. Second, how many genes were testable between each of the comparisons summarised by figure 3? Was this testing done using in a pairwise fashion, using all genes testable between the pair of species being compared (as suggested by line 190), or using a single model matrix with many different contrasts to leverage information across the entire data set? Altough the latter represents a substantial trade off in terms of testable genes, might provide additional insights in terms of polarising results and perhaps even pinpointing the emergence of expression divergence through the tree for specific genes or families. Since subsequent analyses focus primarily on genes identified as DE between human and chimp, I think it would be worth delving a bit more here into the broader temporal trends, or by comparing other interesting pairs of primates.

5. Evolutionary mode of expression levels: as mentioned above, I think the authors do themselves a disservice by not examining their data deeply, eg, by not placing results in a broader evolutionary context or clearly distinguishing between genes that appear to evolve neutrally vs those that exhibit other trends (are there any?). I think this is where the potential of the dataset most shines, and so I strongly encourage the authors to examine the EVEE output in greater detail and see if there's anything interesting hiding in there, although I am cognisant that this might fall outside the scope of the manuscript as they see it, and thus leave it up to them to decide what to do. Nonetheless, some possible questions: can the authors tease out genes evolving under positive selection or showing bursts of accelerated evolution from the overwhelming sea of neutrality? While it's obvious why the authors choose to focus on humans, is anything interesting happening in any other primate lineage?

6. Brain size and positive selection: the authors use mean expression within a species for PCA this time around, as opposed to just all available data points – but it seems to me that this might obscure some trends? Again, the reason for focusing on humans in these sections is obvious – the change in brain size between human and chimp is monumental – but I wonder if this obscures more subtle signals in the data.

*Reviewer #2 (Recommendations for the authors):*

Having commended the authors for compiling this remarkable dataset, I need to share a number of concerns, with regards to data analyses and also interpretation.

1) In general, the analyses and interpretation alternate between investigating general patterns of evolutionary divergence across primate brain regions, and investigating human-specific expression divergence (e.g. Figure 3 is human-based). It may be better to separate the two questions and the analyses used to address them. The authors could start by quantifying the overall phylogenetic signal in brain expression divergence, e.g. using the Chen et al. 2018 model.

In general, I think that performing pairwise DE tests makes little sense with such data (except for cases where specific hypotheses are tested). It would be preferable if the authors studied human-specific expression changes also within an OU-based phylogenetic model that can also incorporate lineage-specific positive selection (e.g. https://doi.org/10.1093/sysbio/syv042).

Also, in analyses about human-specific expression changes, the authors should preferably rely on gene sets which show human-specific upregulation relative to chimpanzees and outgroup species, not just DE gene sets between human and chimpanzee, especially if they wish to interpret the results in the context of other observations (e.g. human-specific positive selection in promoters of semaphorin genes, or increased white matter connectivity).

2) Regarding the analyses on expression patterns correlated with brain size expansions, I would suggest to use some type of phylogenetic residuals analysis, because both brain size and expression will reflect phylogenetic relatedness. So, no surprise that the same genes show correlation across brain regions.

Moreover, it is not clear if the gene list presented in Table 1 and discussed in detail is indeed of statistical significance – multiple testing correction has not been applied.

3) A large number of expression changes may be driven by changes in cell type composition, especially in evolutionary time. At least there should be some discussion on this point in the main text.

Currently the only mention is the supplement, and the content is suboptimal – human brain region bulk transcriptome profiles are compared with cell type-specific transcriptomes of neurons and astrocytes, whereas it is still highly possible that DE genes across species reflect changes in composition. This could be studied explicitly: e.g. https://www.biorxiv.org/content/10.1101/010553v2.

4) Humans are used as reference species in a large number of analyses, but the motivation and rationale behind this is not explained. In fact for questions about general expression divergence in the brain, humans may not be a good reference.

5) Many methodological details are lacking, including crucial information on RNA-seq data quality, how different gene sets were defined, and motivations behind different cutoff choices used, etc. Overall rewriting of methods would be helpful.

[Editors’ note: further revisions were suggested prior to acceptance, as described below.]

Thank you for resubmitting your work entitled "Tempo and mode of gene expression evolution in the brain across Primates" for further consideration by *eLife*. Your revised article has been evaluated by George Perry (Senior Editor) and a Reviewing Editor.

The manuscript has been improved in the revision process, and both reviewers note their appreciation for your responsiveness to the previous round of reviews. I am therefore returning it so that you can address some final issues regarding methodological clarity identified by Reviewer 1, and to give you the opportunity to fix some errors identified by Reviewer 2. In addition, I noticed that even though you have removed the analysis of positive selection on regulatory regions from the manuscript, it is still referenced in your abstract--so please do a thorough read-through for consistency throughout.

*Reviewer #1 (Recommendations for the authors):*

I thank the authors for their reply to my comments and the accompanying revisions and additional details – it's good to see this paper again, as I remain impressed by the dataset and by the scope of the questions the authors are hoping to address.

My responses focus primarily on the points they have addressed in this revision, to avoid dragging this process on forever; I have tried hard to ask only for clarifications, or the absolute minimum, when I have asked for something new. As last time, most of my questions center on methods, so I've brought those to the top, but in all cases, they're requests for additional detail, not for things to be repeated or rerun, so despite the length of the comments I hope they're not too onerous.

1. Trinity output: I would still like a bit more detail on this. Did the authors simply take the best scoring match to a given human gene from the blastnt search? Were there any controls for length or similar? The BUSCO scores are a welcome addition, but some of them are quite low, so additional clarity would be good to help understand what was done. Please note that I am not saying 'redo everything with different thresholds,' but I think more detail would be valuable in parsing the surprisingly low number of genes with data – if 50-70% of the transcriptome is evolving under stabilising selection, but that only covers the 3000 testable genes with one-to-one orthology across the clade, that's actually not a lot of genes at all…

2. PCoA etc: I thank the authors for the additional detail, but I confess I am a bit confused as to how they did what they have done. Line 476 (and 570 for the scRNA data) states that the distance metric is based on log2 fold change distances and that it was generated with plotMDS, but that is not what plotMDS does (plotMDS, by the way, is a limma function, not an edgeR function; limma is loaded in the background when edgeR is loaded). As per the Limma manual, "The distance between each pair of samples (columns) is the root-mean-square deviation (Euclidean distance) for the top genes. Distances on the plot can be interpreted as leading log2-fold-change, meaning the typical (root-mean-square) log2-fold-change between the samples for the genes that distinguish those samples."

If this is not what was done and the authors used a different approach to calculating the distance matrix, why was this done, and how should it be interpreted, especially in light of figure 2? Are the trends reported in this figure (and associated supplements) contingent on the distance metric choice?

3. DE testing methods: It's not clear from the text whether any additional covariates (eg known covariates such as sex or age of the individuals, batches even though they were randomised,) were included in the model or whether samples from the same individual were treated as random effects. As above, I am not asking for a redo at this point, but I think it is important to state these details clearly to ensure reproducibility.

4. Outlier detection: It's not clear to me how this works, from the text (especially as it suggests the opposite of that is described in the response to my original comment). The authors first calculate a mean and sd, and then choose humans as a reference species against which to make comparisons. Does this mean that an outlier gene between humans and chimps is an outlier in the chimpanzee lineage? How can it be an outlier in the human lineage if the human is the reference? I do really appreciate the authors for discussing (line 351) the implications of using humans as the reference, given the general evolutionary shifts we naively expect in the human brain, but in that case, can the data please be included as a supplement?

Other questions:

1. Line 182: Since hominoid expression levels are driving so much of the variation in the PCoA analyses, why do the authors think that this isn't showing up in the phenograms?

2. Line 224: Why are the differences between human and siamang treated as human-specific, rather than informative about all great apes? I do not see how this can be disambiguated with the current setup…

3. Figure 3: I think an upset plot or similar here could be an effective way of visualising results, in terms of exploring how far down the phylogeny some signals are shared, but this is optional.

4. Discussion: I think the sentences beginning in 404 and 407 are fundamentally contradictory, and only 407 seems to truly reflect the results above. Is a reference or a word missing from 404?

---

## [Author Response]

Essential revisions:1) Control for phylogenetic structure in analyzing gene expression changes across the phylogeny, especially in association with brain size. Strongly consider whether phylogenetically informed analysis methods provide greater insight (and reduce the potential for confounding) over pairwise comparisons, especially given advantages afforded by broad sampling across the primate order.

We have recalculated DEGs across brain regions in multiple pairwise comparisons using the OU model and EVEE package (Chen 2018). We added these new analyses to the text (starting at line 189) and have updated Figure 3 with these numbers. We also added information in the methods section on the package used and the specific methods.

2) Provide key missing detail on criteria for defining orthologues; demonstrate robustness of results to decisions about cut-offs (e.g., for defining highly variable genes) and cell type compositional heterogeneity between species.

We have put in a BUSCO table and much more extensive methods sections concerning transcriptome builds and ortholog cutoffs.

The reviewer raises an important point regarding the interpretation of bulk tissue RNA-Seq data in the context of cell type compositional heterogeneity across species. We agree that cell type differences can introduce confounding factors and potentially impact the interpretation of gene expression patterns. However, as we also say to the reviewers that brought up this concern:

Prior knowledge of variation across primates in cell type composition of the brain is informative in interpreting bulk RNA-seq data. It is well known that neuron densities tend to decrease as brain size increases (Sherwood et al., 2020). This suggests that larger brains accommodate a smaller number of neurons per unit volume compared to smaller brains. However, it is interesting to note that other structural elements, such as astrocytes (Munger et al., 2022), microglia (Dos Santos et al., 2020), and synapses (Sherwood et al., 2020), exhibit a relatively invariant density per unit volume across species. Despite changes in brain size, these essential components involved in neural communication and support maintain a consistent presence, emphasizing their crucial role in brain function regardless of the species' brain size. Thus, based on this previous research, in cross-species comparisons of bulk tissue, we can tentatively interpret differences in gene expression to reflect generally similar proportions of major cell types, except for neurons, which are expected to decrease in proportion with larger brain sizes.

Furthermore, it is important to note that while bulk tissue RNA-seq has limitations in capturing cell type-specific gene expression patterns, it can still provide valuable insights into the overall gene expression landscape. By comparing bulk transcriptomes across species, we can identify conserved or divergent expression patterns that may highlight important biological processes or evolutionary trends. Additionally, our study serves as a starting point, providing a foundation for future investigations that can employ more advanced techniques, such as single-cell RNA-seq, to delve deeper into cell type-specific expression patterns.

3) Address methodological questions raised by the reviewers regarding multiple hypothesis testing correction and enrichment analyses.

We have chosen to remove the section looking at positive selection in putative promoter regions from the manuscript and the corresponding table.

4) Test human-specific changes in expression in relationship to outgroup species as well as chimpanzee.

We thank the reviewers for this suggestion, and agree that analyses in the context of outgroup species are vital to understanding what is truly unique to humans. To explore this issue, we took Human and Chimpanzee PFC data from EVEE-tools and ran an enrichment analysis. In the EVEE-tools package, based on the Ornstein-Uhlenbeck model of trait evolution, we can look at expression in human and chimpanzee tissue relative to the rest of the dataset, meaning that all other species in the phylogeny are used as an outgroup. With this, the human PFC was enriched for categories related to “neural growth and development”, “metabolic processes”, and “gene regulation” compared to the chimpanzee PFC in the context of the OU model (as opposed to a null model of random drift). We find that both the number of DEGs as well as the specific categorical enrichments are similar using both EVEE and EdgeR for DE analyses, which ultimately suggests that these changes are relevant in the context of an outgroup species. With this, we are confident that these human-specific changes are representative of the phylogeny presented.

We also looked at Human and Siamang PFC data to compare with an outgroup sample, and the same general trends of enrichment are true in this comparison as well, with enrichments for “Neural Growth/Development”, “Gene Regulation”, and “Metabolic Processes”. This supports the idea that this is human-specific and not just a reflection of changes on the chimpanzee branch.

Reviewer #1 (Recommendations for the authors):1. Trinity output is extremely noisy and returns many isoforms with poor support, or virtually undistinguishable from each other except for a couple of base pairs here and there, especially when run in a de novo mode. How did the authors prioritise the many isoforms, and determine which are credible and worth analysing further? Similarly, since many of the primates lack a reference genome, how did the authors define the set of 3432 testable one-on-one isoforms, or the 15017 genes testable across hominoids? Is it simply on the basis of pairwise orthology to human using BLAST or…? Did the authors control for gene duplication (eg reciprocal BLAST)? In addition, the thresholds for orthology seem very permissive, on the basis of overall coding sequence conservation amongst mammalian species (eg see supp Figure 1B of Chen et al. 2019, where mean coding sequence identity between humans and non-primate one-to-one orthologs is ~85%).

For this and other reviewer questions we have substantially added more details to our Methods section (starting on pg. 12) and moved up more details from the supplement. The testable one-to-one orthologs were taken from the Ensembl database of one-to-one orthologs. For the many species that have published genomes, we also mapped directly to the genome and compared our results to the ones presented here (they were very similar, but more genes are included when mapping directly to the genome, as expected). The thresholds were modeled on another primate transcriptome study (Perry et al. 2012). We have also added BUSCO scores in the Materials and methods (Table S3). The Saki Monkey is the least complete, but we don’t have any results or discussion about that specific lineage and it wasn’t an outlier in any downstream analyses.

2. Many conclusions are based on data from the top 500 most variable genes (generally defined on the basis of SD, but not always – the sanity check against scRNA from astrocytes and neurons uses CoV). Why 500? How robust are all of the results presented to this choice of threshold? Is there a particular species driving this variance (on the basis of PCoA results I dare speculate it's chimpanzees and humans)? How do these observations (PCoA, phenograms) compare to those made on the entire dataset of 3432 testable genes across the entire dataset? It seems to me that the number of genes in the whole data set is not so large as to be worth focusing only on an arbitrarily chosen threshold, and that it would be more informative to consider the entire dataset in these analyses. Similarly, when looking at positive selection, the authors only focus on the top 200 genes DE between human and chimpanzee. Why these limitations?

The top 500 genes is the default state for edgeR and is typical of PCA figures using that package. We also ran the analysis with all of the genes and saw the same patterns. We just show the default for simplicity. The positive selection section has been removed.

3. The authors test the possibility that compositional heterogenity across samples may be biasing their results (line 723 onwards), which I commend, but nonetheless find somewhat incomplete as it currently stands. The only test performed is a comparison of human bulk RNA samples to a small number of astrocyte and neuron RNA-seq samples, which they say proves their tissues are not biased towards either cell type. But combining bulk RNA and scRNA data is not trivial, and since the PCoA shows samples clustering by technology/study (I presume the outlier neuron comes from the same dataset as the astroctyes), it's unclear how to interpret it. Regardless, there's no mention of what I think is the more interesting source of heterogeneity: is there an expectation that the composition of the same tissue type would vary substantially between species? e.g., an excess of, say, glia in the PFC of humans relative to lemurs or loris that could skew results (totally hypothetical example)? I am not sure if this is possible to taste with existing datasets, but it seems to at least be worth discussing?

We agree that mixing the bulk and scRNA is tricky and that there can be confounders. We think that the best approach is to look at the extensive histological literature in primates from the brain.

Prior knowledge of variation across primates in cell type composition of the brain is informative in interpreting bulk RNA-seq data. It is well known that neuron densities tend to decrease as brain size increases (Sherwood et al., 2020). This suggests that larger brains accommodate a smaller number of neurons per unit volume compared to smaller brains. However, it is interesting to note that other structural elements, such as astrocytes (Munger et al., 2022), microglia (Dos Santos et al., 2020), and synapses (Sherwood et al., 2020), exhibit a relatively invariant density per unit volume across species. Despite changes in brain size, these essential components involved in neural communication and support maintain a consistent presence, emphasizing their crucial role in brain function regardless of the species' brain size. Thus, based on this previous research, in cross-species comparisons of bulk tissue, we can tentatively interpret differences in gene expression to reflect generally similar proportions of major cell types, except for neurons, which are expected to decrease in proportion with larger brain sizes.

4. DE testing: As above, I would like to see more detail in the methods here to better contextualise the results. First, I do not think CPM is the accurate unit for comparison here, since it does not control for differences in gene length between species, which may be substantial at some of the orthology thresholds set by the authors – RPKM might be better suited. Second, how many genes were testable between each of the comparisons summarised by figure 3? Was this testing done using in a pairwise fashion, using all genes testable between the pair of species being compared (as suggested by line 190), or using a single model matrix with many different contrasts to leverage information across the entire data set? Altough the latter represents a substantial trade off in terms of testable genes, might provide additional insights in terms of polarising results and perhaps even pinpointing the emergence of expression divergence through the tree for specific genes or families. Since subsequent analyses focus primarily on genes identified as DE between human and chimp, I think it would be worth delving a bit more here into the broader temporal trends, or by comparing other interesting pairs of primates.

Figure 3 was computed using a single model matrix with many different pairwise contrasts. Due to this, the figure does represent fewer testable genes (only 3400 total), but as mentioned above we believe this provides us with the most interesting and relevant points of comparison within the space of this dataset.

We thank the reviewer for the suggestions to look into broader temporal trends, as our expansive dataset allows us the unique ability to better understand some of the more evolutionarily distant primate relationships. In addition to the hominoid comparisons that are the primary focus of the text, we also look at additional species and clade comparisons. Many of these comparisons are shown in Figure 3 and details on the enriched genes in these broader comparisons have been added to the text (lines 250-283).

Significantly, we did not compare all of the species in our dataset in a pairwise manner. This would mean 153 separate differential expression dataset analyses for the whole brain data, and 612 total if we were to analyze on a tissue-specific level. This type of analysis is not within the scope of this paper, and as a result we limited our analyses to only a select few comparisons of interest. Namely, those referenced in the paper for gene categorical enrichment analyses were selected based on the interesting number of differentially expressed genes produced from the EVEE output. However, these other species comparisons can be pursued in future projects and publications.

5. Evolutionary mode of expression levels: as mentioned above, I think the authors do themselves a disservice by not examining their data deeply, eg, by not placing results in a broader evolutionary context or clearly distinguishing between genes that appear to evolve neutrally vs those that exhibit other trends (are there any?). I think this is where the potential of the dataset most shines, and so I strongly encourage the authors to examine the EVEE output in greater detail and see if there's anything interesting hiding in there, although I am cognisant that this might fall outside the scope of the manuscript as they see it, and thus leave it up to them to decide what to do. Nonetheless, some possible questions: can the authors tease out genes evolving under positive selection or showing bursts of accelerated evolution from the overwhelming sea of neutrality? While it's obvious why the authors choose to focus on humans, is anything interesting happening in any other primate lineage?

We thank the reviewer for this comment. We analyzed our dataset for evidence of directional selection and accelerated evolution through a combination of methods that have been added to the manuscript.

Through the EVEE package from Chen 2018, we analyzed the evolutionary history of this dataset through the Ornstein-Uhlenbeck process of continuous trait evolution. With this, we were able to distinguish between gene expression evolving under neutral, stabilizing, and directional selection. We note the percentage of genes that appear to be under stabilizing selection (a majority of the genes in our dataset) (lines 312-326). In contrast to that, we also analyze gene sets that appear to be under directional selection, and thus may be of particular interest to our understanding of primate brain evolution. With this, we utilized two aspects of the EVEE toolset: the ability to score genes for outlier expression (scoreGenes.R) and the ability to identify ‘Differential Expression’ across the phylogenetic tree (ouRegimes.R).

Although the outlier expression script was originally proposed as a method to analyze clinical data, we utilized it here in order to determine the genes that appeared to be the most divergent in terms of expression within our dataset. With this, we expect that genes having significant divergence from the evolutionary mean level of expression and variance (calculated through the EVEE package) would be those that are also under directional selection pressures. We compared our reference sample (human) expression data to that of a test sample, which, in this case, was the mean expression for each individual species comparison. Again, we did not analyze outlier data in all species comparisons across all tissue regions. Rather, we limited this analysis to specific species comparisons of interest. These methods are explained in the manuscript from lines 546-554.

Identification of Differential Expression using the EVEE model employs a multivariate OU model in order to account for multiple regimes of selection (Chen 2018). The methods of this particular model and script are detailed in the manuscript (lines 529-544). With this, we considered “differentially expressed” genes that showed consistent directional shifts to be those likely under directional selection pressures, and thus, evolving under interesting trends (compared to neutral drift, which represented the null hypothesis for this analysis).

The results from the outlier analysis are mentioned in lines 328 to 362, and the results from the differential expression analysis using multiple regimes of selection are detailed from lines 203-283 as well as in Figure 3.

6. Brain size and positive selection: the authors use mean expression within a species for PCA this time around, as opposed to just all available data points – but it seems to me that this might obscure some trends? Again, the reason for focusing on humans in these sections is obvious – the change in brain size between human and chimp is monumental – but I wonder if this obscures more subtle signals in the data.

We kind of have to use ECVs when dealing with this wide of a range of primates since brain size data doesn’t exist that we know of. We’ve added to the methods that we added a human ECV (also averaged of male and female) from data published by (Coqueugniot and Hublin, 2012; https://doi.org/10.1002/ajpa.21655). Isler et al. (2008) reported minimal error when ECV was transformed to brain mass using the correction factor of 1.036g/cc (Stephan, 1960). WE transformed all the ECV to brain sizes using this conversion. This way we can truly say that we were looking for correlations with brain size and not ECV.

Reviewer #2 (Recommendations for the authors):Having commended the authors for compiling this remarkable dataset, I need to share a number of concerns, with regards to data analyses and also interpretation.1) In general, the analyses and interpretation alternate between investigating general patterns of evolutionary divergence across primate brain regions, and investigating human-specific expression divergence (e.g. Figure 3 is human-based). It may be better to separate the two questions and the analyses used to address them. The authors could start by quantifying the overall phylogenetic signal in brain expression divergence, e.g. using the Chen et al. 2018 model.

Thank you very much for the suggestion of the model from Chen 2018.

We used the EVEE toolset as a method to look at both tissue-specific expression divergences, as well as for quantification of the overall phylogenetic signal in the brain in regards to this comment.

In the context of the OU model, the overall phylogenetic signal in brain expression divergence was slightly smaller than was observed with EdgeR. This is likely a reflection of the fact that this model is looking at constrained trait expression evolution in the context of phylogenetic relationships, which represents a more stringent type of analysis.

In general, I think that performing pairwise DE tests makes little sense with such data (except for cases where specific hypotheses are tested). It would be preferable if the authors studied human-specific expression changes also within an OU-based phylogenetic model that can also incorporate lineage-specific positive selection (e.g. https://doi.org/10.1093/sysbio/syv042).

As we have noted in other reviewer comments we did DE in the context of the OU model and have included those numbers in the text and have remade Figure 3 with those numbers.

Also, in analyses about human-specific expression changes, the authors should preferably rely on gene sets which show human-specific upregulation relative to chimpanzees and outgroup species, not just DE gene sets between human and chimpanzee, especially if they wish to interpret the results in the context of other observations (e.g. human-specific positive selection in promoters of semaphorin genes, or increased white matter connectivity).

We agree that human-specific DE trends should be analyzed in reference to multiple outgroup species, and we have addressed this point above in a previous reviewer comment. We have also revised statements made about selection in semaphorin genes to look at more broad species comparisons, and these are now in the manuscript.

2) Regarding the analyses on expression patterns correlated with brain size expansions, I would suggest to use some type of phylogenetic residuals analysis, because both brain size and expression will reflect phylogenetic relatedness. So, no surprise that the same genes show correlation across brain regions.Moreover, it is not clear if the gene list presented in Table 1 and discussed in detail is indeed of statistical significance – multiple testing correction has not been applied.

Based on this concern and those listed below, we have chosen to remove this second section of the manuscript and the table. We address the first part where it’s repeated below.

3) A large number of expression changes may be driven by changes in cell type composition, especially in evolutionary time. At least there should be some discussion on this point in the main text.Currently the only mention is the supplement, and the content is suboptimal – human brain region bulk transcriptome profiles are compared with cell type-specific transcriptomes of neurons and astrocytes, whereas it is still highly possible that DE genes across species reflect changes in composition. This could be studied explicitly: e.g. https://www.biorxiv.org/content/10.1101/010553v2.

We’ve added a large section describing what was already known about this potential issue to that part of the methods section.

4) Humans are used as reference species in a large number of analyses, but the motivation and rationale behind this is not explained. In fact for questions about general expression divergence in the brain, humans may not be a good reference.

We thank the reviewer for this perspective. In order to investigate this we repeated all of our EVEE-based analyses using two other reference species, siamang and rhesus macaque. In this type of analysis, the reference species refers to a select subset of our entire dataset with which we compare expression differences to. This is used in order to normalize counts in our dataset for downstream analysis.

In the manuscript, we report the percentage of genes that fit better under the model of stabilizing selection as compared to neutral drift. These numbers are not significantly different from the human comparison data and ultimately, a majority of genes still fit better under the OU model. This information has been added at lines 364-387.

We also looked at select pairwise species comparisons with these alternate reference species in order to determine if the general trends of directional selection and DGE were also similar to the human reference dataset. We again saw similar numbers of DEGs, and the enrichment categories of those significant terms also followed the same trends no matter what reference species was used in the analysis. The only terms that would differ would be those that had the highest p-values and thus were the least statistically significant. We have added this to the manuscript at lines 364-387.

5) Many methodological details are lacking, including crucial information on RNA-seq data quality, how different gene sets were defined, and motivations behind different cutoff choices used, etc. Overall rewriting of methods would be helpful.

We agree and have now extensively rewritten and expanded the methods section (starting on pg 12).

[Editors’ note: what follows is the authors’ response to the second round of review.]

The manuscript has been improved in the revision process, and both reviewers note their appreciation for your responsiveness to the previous round of reviews. I am therefore returning it so that you can address some final issues regarding methodological clarity identified by Reviewer 1. In addition, I noticed that even though you have removed the analysis of positive selection on regulatory regions from the manuscript, it is still referenced in your abstract--so please do a thorough read-through for consistency throughout.

We have addressed the reviewers’ comments and have added a optional figure (an Upset Plot now Figure 3 – supplemental figure 2) and a table looking at the effects of which species is chosen as the outlier (Supplemental Table 10). We’re grateful for these thoughtful reviewer comments. Thank you for the catch on the abstract!

Reviewer #1 (Recommendations for the authors):I thank the authors for their reply to my comments and the accompanying revisions and additional details – it's good to see this paper again, as I remain impressed by the dataset and by the scope of the questions the authors are hoping to address.My responses focus primarily on the points they have addressed in this revision, to avoid dragging this process on forever; I have tried hard to ask only for clarifications, or the absolute minimum, when I have asked for something new. As last time, most of my questions center on methods, so I've brought those to the top, but in all cases, they're requests for additional detail, not for things to be repeated or rerun, so despite the length of the comments I hope they're not too onerous.1. Trinity output: I would still like a bit more detail on this. Did the authors simply take the best scoring match to a given human gene from the blastnt search? Were there any controls for length or similar? The BUSCO scores are a welcome addition, but some of them are quite low, so additional clarity would be good to help understand what was done. Please note that I am not saying 'redo everything with different thresholds,' but I think more detail would be valuable in parsing the surprisingly low number of genes with data – if 50-70% of the transcriptome is evolving under stabilising selection, but that only covers the 3000 testable genes with one-to-one orthology across the clade, that's actually not a lot of genes at all…

We thank the reviewers for this comment and are happy to provide more explanation on our Trinity assemblies (as added to the methods section, lines 494-497). With Trinity assembly, fragments of the original RNA reads for each species are compiled and clustered into groups based on sequence similarity, which eventually are extended to reconstruct full-length transcripts. Following Trinity assembly, we mapped our individual samples to each assembled transcriptome using Bowtie2, and from there we further process samples using HTSeq to assign read counts to annotated genomic features (using a GTF file obtained from Ensembl). With this, the BLAST database is never used to directly map to the human genome. Additionally, there are no specific controls for length of transcripts.

Regarding the lower range of BUSCO scores, we would like to highlight the fact that the dataset only consists of brain tissue. Previous studies, including the original BUSCO publication, show that in general transcriptome assemblies from single tissue regions on average have lower completeness scores than assemblies composed of reads from a variety of tissue types (Simao 2015). This is likely a reflection of tissue-specific gene expression. As evidence of this, we further analyzed reads deemed as “missing” by BUSCO and found that many of these showed little to no expression across the human brain (The Human Protein Atlas; Uhlen 2015). This information has been added to the Methods section (lines 505-511).

We would also like to note that the species with the lowest BUSCO scores (and therefore the least complete transcriptomes) are the Slow Loris and Slender Loris. We do not make any major conclusions about the Loris species in this manuscript and thus do not believe these BUSCO scores will be majorly affecting the conclusions made in this manuscript. We only consider Loris data in concert with Lemurs, which by comparison have much more complete transcriptomes (comparable to other primate brain datasets; Simao 2015). This information has been added to the Methods section (lines 511-516).

We are limited to roughly 3000 genes for analysis in order to compare one-to-one orthologs across 18 primates and 70-90 million years of evolutionary history. We recognize that this is a significant limitation and that we are missing a large number of genes in this dataset that may be important to brain evolution on a smaller timescale. However, in order to better characterize long-term primate evolution in these particular brain regions, we believe that the addition of more evolutionary divergent species enhances the significance of the data presented.

2. PCoA etc: I thank the authors for the additional detail, but I confess I am a bit confused as to how they did what they have done. Line 476 (and 570 for the scRNA data) states that the distance metric is based on log2 fold change distances and that it was generated with plotMDS, but that is not what plotMDS does (plotMDS, by the way, is a limma function, not an edgeR function; limma is loaded in the background when edgeR is loaded). As per the Limma manual, "The distance between each pair of samples (columns) is the root-mean-square deviation (Euclidean distance) for the top genes. Distances on the plot can be interpreted as leading log2-fold-change, meaning the typical (root-mean-square) log2-fold-change between the samples for the genes that distinguish those samples."If this is not what was done and the authors used a different approach to calculating the distance matrix, why was this done, and how should it be interpreted, especially in light of figure 2? Are the trends reported in this figure (and associated supplements) contingent on the distance metric choice?

We thank the reviewer for this method clarification and have added this information about plotMDS to the text (lines 521-525). To build the PCoAs, we are utilizing EdgeR’s plotMDS.DGElist function that is designed for gene expression datasets. Within this (as described in the EdgeR manual), gene counts from a DGElist are converted to log-counts-per-million and then passed on to limma where distances are calculated based on leading log fold changes. To define this further, the leading log fold chance is the “average (root-mean-square) of the largest absolute log-fold-changes between each pair of samples”. (Robinson, McCarthy, and Smyth, 2010)

3. DE testing methods: It's not clear from the text whether any additional covariates (eg known covariates such as sex or age of the individuals, batches even though they were randomised,) were included in the model or whether samples from the same individual were treated as random effects. As above, I am not asking for a redo at this point, but I think it is important to state these details clearly to ensure reproducibility.

We agree with the comment above that information on other sources of variation is important to ensure the conclusions made from this dataset are reproducible. To that end, no additional covariates were included in either the EdgeR DE analysis or the EVEE-tools OU-model based DE analysis. However, we did conduct additional ANCOVA covariate testing to confirm that other factors, including age and sex, were not significant sources of variation in gene expression (The median contribution of age and sex to the gene expression data was about 0.3% for both factors). These analyses, along with our PCoA plots, show that species identity (12% contribution) and brain region (6.6% contribution) are the two most significant determinants for DE. Additionally, differences in samples from the same individual were assumed to be mostly linked to brain region, as both region and individual show similar levels of contribution in the ANCOVA data (6.6% and 5% respectively). These ANCOVA analyses did show minor residual effects that we deemed asrandom and were not further analyzed for the purposes of this manuscript.

This information is added to the methods section (lines 590-596).

Author response table 1 is the data from the ANCOVA analysis (a multi-way ANOVA) performed using the lm function from the stats package in EdgeR.

**Author response table 1. sa2table1:** 

term	min	q25	median	mean	q75	max	sd
Residuals	0.01258956	0.23478366	0.31134162	0.31409894	0.38774308	0.75204413	0.11557519
age	8.2272E-10	0.00049112	0.003355	0.01021905	0.01296546	0.19166529	0.01660386
family	0.00018917	0.151815	0.2737052	0.30422055	0.43594555	0.87422117	0.18970219
individual	0.00042591	0.02788199	0.05329402	0.06773906	0.09440077	0.37780784	0.05345294
region	0.00021586	0.03412893	0.06644626	0.10044476	0.12891681	0.88163452	0.10373119
sex	5.0908E-10	0.00052758	0.00313619	0.00908844	0.01174799	0.1501228	0.01403884
species	2.0257E-06	0.04136567	0.12098179	0.1941892	0.3078835	0.8005367	0.18885516

4. Outlier detection: It's not clear to me how this works, from the text (especially as it suggests the opposite of that is described in the response to my original comment). The authors first calculate a mean and sd, and then choose humans as a reference species against which to make comparisons. Does this mean that an outlier gene between humans and chimps is an outlier in the chimpanzee lineage? How can it be an outlier in the human lineage if the human is the reference? I do really appreciate the authors for discussing (line 351) the implications of using humans as the reference, given the general evolutionary shifts we naively expect in the human brain, but in that case, can the data please be included as a supplement?

We thank the reviewers for this comment and have added a more detailed explanation of the methods used for outlier analysis to the text (lines 598-614). First, upon further analysis, we agree that we *cannot* define outliers for species being used as the reference. With this, we have adjusted the conclusions made in this section of the paper to reflect different methods of testing: for conclusions on “human outliers” we are utilizing rhesus macaque as a reference (lines 343-380).

With this revised methodology, we see outliers in the human prefrontal cortex that include disease-relevant genes, such as Amyloid Precursor Protein (APP) (O'Brien and Wong 2011) and Parkin (PRKN) (Funayama et al. 2023). With this, we also see a select number of metabolism relevant genes, WNT signaling regulators, and members of the neurexin family. We also find outliers in the chimpanzee PFC which include many genes implicated in metabolic pathways, such as ANKH (Szeri et al. 2020) and ETFA (Henriques et al. 2021) (ETFA is also found to be an outlier in humans which suggests that its upregulation is shared within the hominini tribe). This new data has been added to the text (lines 343-380), with citations to support the functional information provided.

We'd also like to clarify the text and better define the term “reference” as the species with which expression values across the entire dataset are TMM-normalized to. In this analysis, we tested two different reference species (human and rhesus) depending on which species we were analyzing for outlier expression. We’ve added in the data for both of these analyses to Supplementary Table 10 as well as further clarified the methods (lines 598-614) for this analysis.

Other questions:1. Line 182: Since hominoid expression levels are driving so much of the variation in the PCoA analyses, why do the authors think that this isn't showing up in the phenograms?

We think that’s due to the way the phenograms were constructed. We used neighbor-joining for a simple look at the way gene expression was clustering the samples. Since neighbor-joining looks for a minimum sum of branch lengths, and is somewhat weighted by what pairs are grouped in the initial analysis (Felsenstein 2004), we used it more as a simple way to look at how the different brain regions were clustering and less of a metric than the PCoA.

2. Line 224: Why are the differences between human and siamang treated as human-specific, rather than informative about all great apes? I do not see how this can be disambiguated with the current setup…

We refer to these differences as “human specific” due to the nature of the analysis. With the OU model, we are defining our particular regimes for multivariate analysis (Chen et al., 2018). In this case, our first regime of interest is human and the second regime is siamang. We directly are comparing the gene expression patterns for these species, while all other 16 species in the dataset (including the great apes) are included as “background” and used to infer whether stabilizing selection is the most significant evolutionary model. While it is likely that these differences are informative about all great apes relative to the siamang, we did not conduct an individual analysis of each great ape species to the siamang to make this conclusion.

3. Figure 3: I think an upset plot or similar here could be an effective way of visualising results, in terms of exploring how far down the phylogeny some signals are shared, but this is optional.

We thank the reviewer for this suggestion and have added upset plots, separated out by brain region, as Figure 3 – supplementary figure 2. We also reference these in our analyses to show how in general, the number of differentially expressed genes increases over evolutionary time (lines 200-216; methods lines 568-575). We would like to note, however, the way that these upset plots are constructed (using the UpsetR package in RStudio) does not actually allow us to analyze shared signals as the plot only shows the total number of differentially expressed genes, rather than the identity and functional categories of these genes (I.e. 200 DE genes in one species may not be the same 200 in another species). Rather, we find that the gene ontology enrichment analysis conducted using GProfiler is more useful to look for shared signals, and those analyses are included in this manuscript along with the data presented in Figure 3.

4. Discussion: I think the sentences beginning in 404 and 407 are fundamentally contradictory, and only 407 seems to truly reflect the results above. Is a reference or a word missing from 404?

Thank you for noting that! It has been changed and a reference added to make the contrast with another study looking at all mammals. It is now lines 436-438.